# CrossCheck: A Vision-Language Conflict Detection Benchmark

## Abstract

Contradictory multimodal inputs are common in real-world settings, yet existing benchmarks typically assume input consistency and fail to evaluate cross-modal conflict detection – a fundamental capability for preventing hallucinations and ensuring reliability. We introduce CrossCheck, a novel benchmark for **multimodal conflict detection**, featuring COCO images paired with contradictory captions containing controlled object-level or attribute-level conflicts. Each sample includes targeted questions evaluated in both multiple-choice and open-ended formats. The benchmark provides an extensive fine-tuning set filtered through automated quality checks, alongside a smaller human-verified diagnostic set. Our analysis of state-of-the-art models reveals substantial limitations in recognizing cross-modal contradictions, exposing systematic modality biases and category-specific weaknesses. Furthermore, we empirically demonstrate that targeted fine-tuning on CrossCheck substantially enhances conflict detection capabilities.

## 1 Introduction

Multimodal Large Language Models (MM-LLMs) (Yu et al., 2024; Liu et al., 2023a; Zhu et al., 2023; Liu et al., 2023b; Ye et al., 2023) have achieved remarkable progress in cross-modal understanding, demonstrating impressive capabilities in tasks ranging from image captioning and visual question answering to complex multimodal reasoning tasks. However, real-world applications frequently present these models with contradictory information across modalities: medical systems reporting "no abnormalities" while X-rays show fractures, autonomous vehicles detecting "clear roads" despite camera feeds showing barriers, or financial documents where text and scanned amounts differ. In such critical scenarios, models have to recognize contradictions, reason about information reliability across modalities, and flag inconsistencies for human review. Despite these practical demands, there remains a significant gap in evaluation benchmarks designed to assess MM-LLMs' ability to detect and resolve conflicts between visual and textual information – a fundamental skill required for robust real-world deployment.

Existing benchmarks, while valuable for assessing general multimodal capabilities, fall short in evaluating conflict detection. Visual question answering datasets like VQA (Antol et al., 2015) and OK-VQA (Marino et al., 2019) primarily focus on information extraction and basic reasoning, assuming consistency between visual and textual inputs. Multimodal reasoning benchmarks such as CLEVR (Johnson et al., 2017) and GQA (Hudson & Manning, 2019) emphasize compositional understanding but do not systematically evaluate models across diverse cross-modal conflict scenarios. Recent comprehensive evaluations like MMBench (Liu et al., 2024b) and SEED-Bench (Li et al., 2023a) cover broad multimodal competencies but lack dedicated assessment of conflict detection.

The challenge of multimodal conflict detection encompasses several key difficulties. First, models must possess sufficient **cross-modal reasoning** capabilities to compare and contrast information across modalities rather than processing them independently. Second, robust evaluation requires assessing **modality bias patterns** – understanding whether models systematically favor visual or textual information when faced with conflicts. Third, effective evaluation should span **diverse semantic categories** and **response formats** – testing models on various conflict types (objects, attributes), while accommodating format flexibility from multiple-choice to free-form outputs. Finally, it remains unexplored whether models can **learn to improve** at conflict detection through targeted training, or if observed weaknesses reflect deeper architectural limitations.

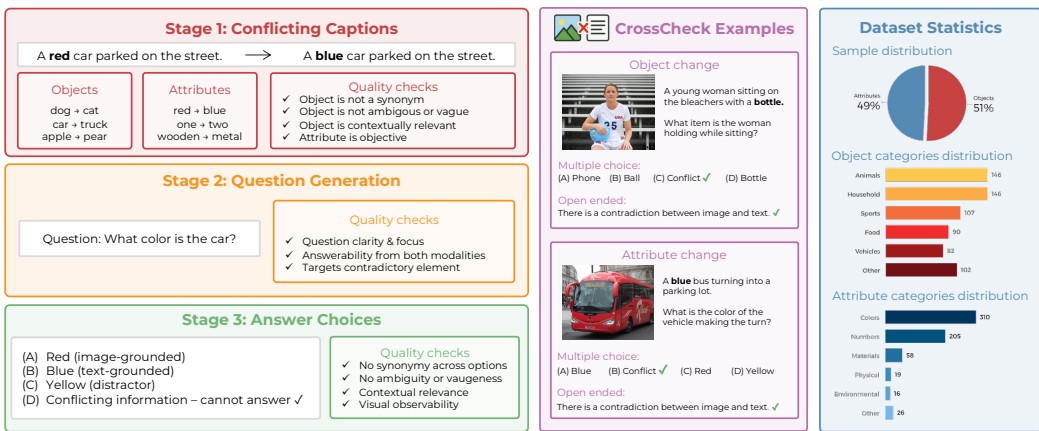

Figure 1: **Left:** Three-stage pipeline generates cconflicting image–text pairs from MS COCO with targeted questions. **Middle:** Examples in CROSSCHECK, showing object and attribute contradictions. Models are evaluated on the ability to detect conflicts in multiple-choice or open-ended format. **Right:** The diagnostic set includes 1289 categorized and human-verified test samples.

In response, we introduce CROSSCHECK, a comprehensive vision-language conflict detection benchmark designed to systematically evaluate MM-LLMs' ability to identify multimodal contradictions. CROSSCHECK features carefully constructed image-text pairs, sourcing images from MS COCO (Lin et al., 2014) and pairing them with contradictory text descriptions across object-level and attribute-level categories. Each sample contains precisely one controlled contradiction between the visual and textual content, accompanied by questions in both multiple-choice and open-ended formats. Our benchmark provides ∼15k high-quality training samples and a human-verified diagnostic set, enabling detailed analysis of model performance and assessment of improvement through targeted fine-tuning. Fig. 1 shows an overview of the generation pipeline and resulting benchmark.

Using CROSSCHECK, we evaluate state-of-the-art models and reveal striking performance disparities: leading closed–source models (GPT-5 (OpenAI, 2025), Gemini 2.5 Pro (Team et al., 2023)) achieve strong performance (>85% accuracy) while most open-ended models struggle with near-zero detection rates. Our analysis not only exposes systematic modality biases and categorical weaknesses, but also crucially demonstrates that fine-tuning can dramatically improve the conflict detection capabilities, *e.g.*, the performance of LLaVa-1.5-7b (Liu et al., 2023b) improves from 0% to 77%. These findings highlight critical gaps in current multimodal systems while establishing targeted fine-tuning as an effective solution for robust conflict detection.

In summary, our key **contributions** are:

① We introduce CROSSCHECK, a benchmark for multimodal conflict detection, with ∼15k training samples and 1289 human-verified test cases across fine-grained object and attribute categories.
② We conduct extensive evaluation of state-of-the-art MM-LLMs, revealing significant performance gaps between closed-source and open-ended models, systematic modality biases, and category-specific weaknesses in conflict detection capabilities.
③ We demonstrate that targeted fine-tuning can dramatically improve conflict detection performance of MM-LLMs, with some models achieving over 75% accuracy improvement.

## 2 RELATED WORK

**Multimodal large language models.** MM-LLMs use powerful LLMs to enable joint reasoning across visual and textual information, demonstrating emergent capabilities like detailed image captioning, visual question answering, and open-ended multimodal dialogue. Closed-source models including GPT (Achiam et al., 2023; OpenAI, 2025) and Gemini (Team et al., 2023) have established strong baselines for multimodal reasoning across diverse benchmarks. In parallel, the open-source community has developed a diverse ecosystem of multimodal models, adopting different designs for processing visual and textual information. Examples include instruction-tuned systems

like InstructBLIP (Dai et al., 2023), LLaVA (Liu et al., 2024a), and MiniGPT4 (Zhu et al., 2023), as well as scaling-focused efforts such as InternVL (Chen et al., 2024b), Qwen-VL (Wang et al., 2024a), Phi3-Vision (Abdin et al., 2024), and mPLUG-Owl (Ye et al., 2023). These models vary in architectural design (encoder-decoder vs. decoder-only), training objectives (instruction tuning vs. alignment with human feedback), and supervision signals (synthetic vs. human-curated).

Despite the progress in MM-LLMs, there exist some challenges that undermine their performance and reliability: i) *hallucinations*, where models describe non-existing objects or attributes (Leng et al., 2024; Chen et al., 2024a), ii) *modality bias*, where the autoregressive nature of LLM leads to an over-reliance on language priors (Zhu et al., 2025; An et al., 2025) and a progressive visual dilution in long responses (Wang et al., 2025; Yang et al., 2025), and iii) *limited interpretability of internal representations*, as current models lack transparency in how visual and textual information is processed and integrated within their latent spaces (Jiang et al., 2025). These issues highlight the need for diagnostic tools to assess model behavior when visual and linguistic inputs diverge.

**Multimodal benchmarks.** General-purpose benchmarks such as MME (Fu et al., 2023), MMBench (Liu et al., 2024b), and SEED-Bench (Li et al., 2023a) evaluate broad multimodal capabilities but assume consistent inputs, while most reliability studies focus on hallucinations rather than explicit conflict detection. Early benchmarks introduced simple Yes/No questions, *e.g.*, POPE (Li et al., 2023b) for object hallucination evaluation and NOPE (Lovenia et al., 2023) for negative object presence. Recent work has expanded hallucination taxonomy and evaluation methods. HaloQuest (Wang et al., 2024b) introduced "false premise" questions about non-existing objects or attributes, while LRV (Liu et al., 2023a) proposed "negative instruction" with nonexistent object manipulation. AutoHallusion (Wu et al., 2024) modifies image contents using DALLE-2/3 for object insertion and removal to create mismatches. Similarly, Koala (Carragher et al., 2025) studies knowledge conflicts by applying targeted perturbations to image sources. However, their inpainting and editing operations rely on pretrained models (Suvorov et al., 2022; Rombach et al., 2022) and often lead to unrealistic images. HallusionBench (Guan et al., 2024) provides a diagnostic suite for analyzing multimodal failures, revealing that models suffer from both language hallucination (prioritizing prior knowledge over visual context) and visual illusion (producing incorrect answers about given figures). MMKC-Bench (Jia et al., 2025) investigates knowledge conflicts in a retrieval-augmented generation framework. Specialized efforts in multimodal misinformation detection include TRUST-VL, which addresses textual, visual, and cross-modal distortions in news content (Yan et al., 2025).

Unlike prior work, CROSSCHECK is as a diagnostic benchmark designed for *multimodal conflict detection*. It systematically introduces controlled contradictions into image–text pairs and evaluates models across both multiple-choice and open-ended question answering formats.

## 3 🖼️❌📋 CROSSCHECK: CROSS-MODAL CONFLICT DETECTION BENCHMARK

CROSSCHECK evaluates MM-LLMs' ability to detect inconsistencies between visual and textual inputs. Unlike traditional benchmarks that assume modality consistency or designate one input as ground truth, CROSSCHECK requires models to peform cross-modal reasoning and identify contradictions – reflecting real-world scenarios where either modality may contain errors or hallucinations.

### 3.1 DATASET OVERVIEW

Each sample in CROSSCHECK consists of an **image** from MS COCO (Lin et al., 2014), a **text** description that contradicts the visual content in one aspect, and a targeted **question** about the contradictory element. The contradictions span two categories: **object-level** and **attribute-level** conflicts.

Our benchmark includes two complementary evaluation tasks: (i) **multiple-choice question answering**, with four carefully desgined options – image-grounded, text-grounded, plausible distractor, and "Conflicting information – cannot answer" (the correct choice), and (ii) **open-ended question answering**, where models generate free-form responses.

CROSSCHECK is split into a **training set** (∼15k high-quality samples filtered from an initial ∼30k generated samples), and a human-verified **test set**, which serves as a diagnostic benchmark. Test samples cover 655 object-level and 634 attribute-level contradictions. Figure 2 shows the distribution of object and attribute categories, and question types in the diagnostic subset.

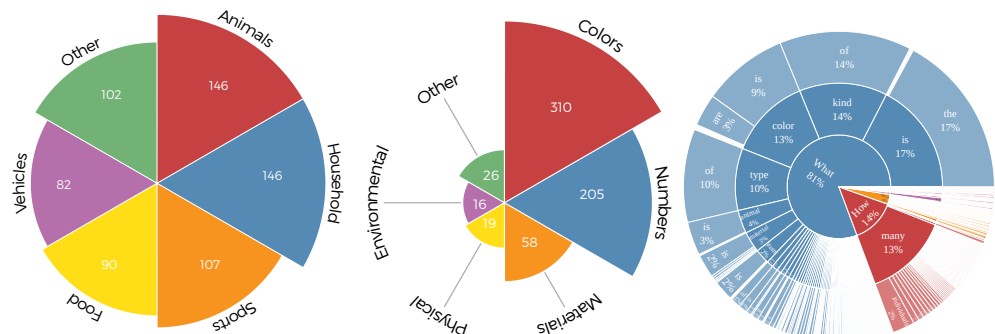

Figure 2: Diagnostic set statistics. **Left:** Object cateogory distribution (655 samples). **Middle:** Attribute distribution (634 samples). **Right:** Distribution of questions by their first three words.

## 3.2 DATASET CONSTRUCTION PIPELINE

The dataset is built through a multi-stage pipeline, using an LLM (Gemini 2.5 Flash) to generate contradictions, questions, and answer choices (prompts in App. B.1). Each stage includes automatic quality filters, with human verification of the final test set.

**Stage 1: Conflicting caption generation.** Given an original COCO caption, an LLM modifies exactly one element – an object or an objective attribute (*e.g.*, color, number, shape, material) – while keeping the rest unchanged. The system tracks the *changed words*, categorizes the *change type* (object vs. attribute), and ensures the modification creates a plausible but incorrect alternative.

**Stage 2: Question generation.** Based on the conflicting caption, an LLM generates subtle questions that focus on the contradictory element without explicitly indicating an error. Questions are designed to be answerable from both the original and conflicting captions, but with different answers.

**Stage 3: Answer generation.** For multiple-choice questions, three distinct answers are created: image-grounded (based on the original caption), text-grounded (based on the conflicting caption), and a contextually plausible distractor that appears in neither caption.

## 3.3 QUALITY CONTROLS

Given the synthetic nature of our dataset, we implement a multi-stage quality control framework that combines automated validation at each generation step with final human oversight of the test set to ensure reliability. Our automated validation includes both rule-based checks and LLM-based assessment using Gemini 2.5 Flash Lite, with prompts given in App. B.2.

**Caption editing validation.** We validate caption modifications through two complementary approaches. *Automated checks* verify basic correctness through word-level validation: (i) the original word must appear in the source caption, and (ii) the conflicting word must appear in the modified caption. This prevents hallucinated changes that don't correspond to recorded modifications. *LLM-based validation* enusres semantic consistency of modifications. For attribute changes, we verify that original and conflicting words are indeed attributes and classify them as objective (measurable, factual properties like: red, square, wooden) vs. subjective (opinion-based descriptors like: beautiful, large, elegant). For object changes, we make sure both words are objects, verify they are not synonyms or ambiguous terms, and ensure contextual relevance.

**Question generation validation.** We ensure generated questions meet three criteria: clarity, focus, and answerability. Our validation identifies ambiguous phrasing, verifies questions target the modified elements rather than irrelevant aspects, and confirms answerability by ensuring all candidate answers are semantically compatible with the question, preventing type mismatches.

**Answer generation validation.** We combine automated checks with LLM-based assessment for the generated answers. *Automated validation* verifies answer consistency: (i) the image-only answer matches the original word (ii) the text-only answer corresponds to the conflicting word and (iii) the distractor answer appears in neither caption. *LLM-based validation* ensures answer quality

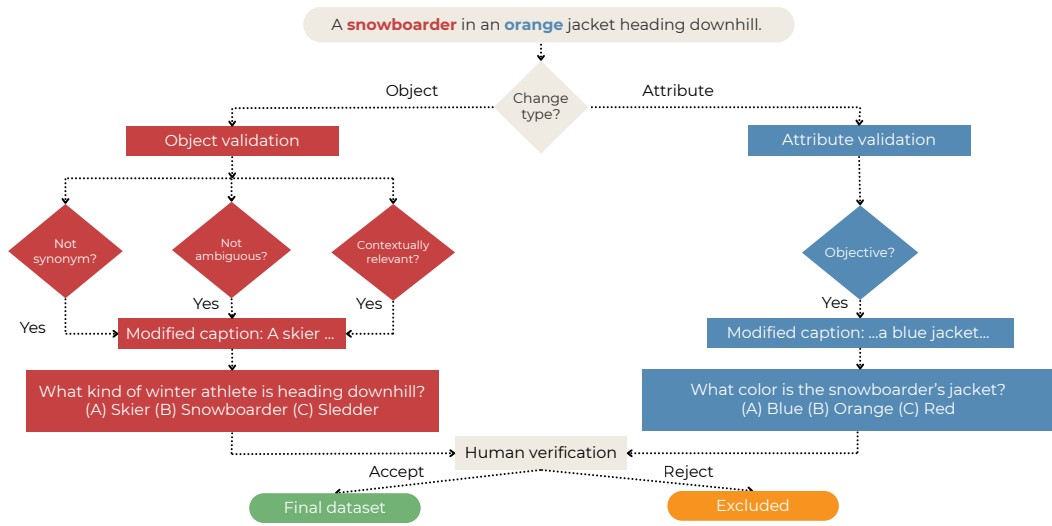

Figure 3: Dataset generation pipeline. Starting from MS COCO captions, the pipeline identifies change type (object vs. attribute) and applies corresponding validation checks. Validated changes proceed to question generation before human verification determines final dataset acceptance. Examples show object change (snowboarder → skier) and attribute change (orange → blue).

by checking for no near-duplicate or synonymous answers, identifies ambiguous terms ("several", "medium"), assesses contextual relevance, and verifies that all answers represent directly observable visual concepts rather than abstract properties.

**Human verification.** To ensure a high-quality evaluation benchmark and validate our automated quality control, we conduct human evaluation on the filtered test set. Annotators follow structured guidelines that reflect our automated validation criteria. Instructions and examples of accepted and rejected samples are provided in App. G. Each test sample receives a binary accept/reject vote from a human annotator, and only samples marked as accepted are included in the final dataset.

In summary, the systematic dataset construction and multi-stage quality controls produce a reliable collection of image–text contradictions, with human-verified test samples ensuring benchmark validity. The full construction and validation pipeline is illustrated in Fig. 3.

## 3.4 DATASET CATEGORIZATION METHODOLOGY

Following the dataset generation and quality assurance procedures, we organized the data into categories to support fine-grained evaluation beyond overall accuracy. To systematically define object and attribute categories in CROSSCHECK, we combine frequency analysis with manual curation.

**Frequency analysis.** We extracted the transformed word pairs from each dataset sample (e.g., "dog" → "cat"), computed frequency distributions by change type (object vs. attribute), and identified the 20 most common terms in each category. See Fig. 4 for visualization of the most common words.

**Category definition.** Based on the frequency analysis, we defined five object categories (animals, vehicles, sports, food, furniture) and five attribute categories (color, number, material/texture, physical properties and environmental conditions). We used an LLM (GPT 5 (OpenAI, 2025)) to categorize all changed words into predefined categories, then manually verified the assignments (see App D and E for the complete word categorizations).

**Sample assignment.** Samples were assigned using strict co-membership: both original and conflicting words must belong to the same category, with non-conforming samples assigned to "Other".

Organizing samples by semantic categories isolates specific weaknesses in models' conflict detection abilities and helps to determine whether failures stem from object recognition or attribute understanding. Beyond aggregate metrics, this revealas systematic biases and areas for improvement. See App. F for qualitative examples of each category.

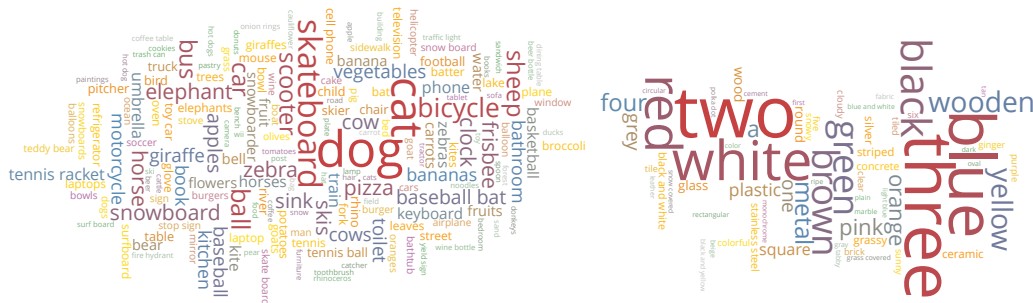

Figure 4: Word frequency analysis. **Left:** Most frequent words appearing in object contradiction pairs. **Right:** Most common words from attribute contradiction pairs.

# 4 RESULTS AND DISCUSSION

## 4.1 EXPERIMENTAL SETUP

**Evaluation metrics.** We report accuracy with bootstrap-estimated standard deviations (1000 iterations). We use strict string matching for multiple-choice and relaxed matching for open-ended responses (see App. A.2). The codebase will be released upon acceptance.

**Models.** We evaluate two model categories on CROSSCHECK. Closed-source models include Gemini (2.5 Pro, 2.5 Flash Lite) and GPT variants (5, 4.1 Mini). Open-source models cover diverse architectures: InstructBLIP (Dai et al., 2023), InternVL 1.5 (Chen et al., 2024b), LLaVa-1.5 (Liu et al., 2024a), MiniGPT4 (Zhu et al., 2023), LRV-MiniGPT4 (Liu et al., 2023a), Qwen-7B (Wang et al., 2024a), Phi3-Vision-128k (Abdin et al., 2024), and mPLUG-Owl (Ye et al., 2023). All models are used following official implementations and provided checkpoints.

## 4.2 MULTIPLE CHOICE QUESTION ANSWERING

We evaluate models on a multiple-choice QA task with four options: "Contradiction" (correct), "Image"-grounded, "Text"-grounded answer, and a "Distractor", randomly assigned to ((A)-(D)). This design allows us to measure both conflict detection accuracy and modality bias patterns across different model architectures.

**Evaluation protocol.** Multiple-choice responses use strict string matching ((A)–(D)), with mismatches labeled "Incorrect" to capture instruction-following and reasoning errors. Results using relaxed string matching (App. A.1) are reported in Table 5 in App. C.

**Performance evaluation.** Table 1 presents the distribution of model predictions on CROSS-CHECK across answer choices [1]. The results reveal three key findings, discussed below.

*(i) Performance hierarchy.* The results demonstrate a clear performance divide between closed-source and open-ended models. Top-tier closed-source models (GPT 5 and Gemini 2.5 Pro) demonstrate strong conflict detection capabilities, correctly identifying conflicts 86.78% and 88.48% of the time, respectively. Their strong performance also validates that our dataset contains well-constructed, unambiguous samples with clear correct answers and confirms the task is solvable given sufficient reasoning capabilities. In contrast, most open-ended models struggle significantly, with many achieving less than 3% accuracy on the primary task, revealing a substantial capability gap in multimodal conflict reasoning. Notably, InstructBLIP-T5-xxl performs best among open-ended models with 63.87% contradiction detection, though still far below the top-tier models.

*(ii) Modality bias patterns.* When models fail to detect contradictions, they exhibit distinct bias patterns. Specifically, GPT-4.1 Mini and Gemini 2.5 Flash Lite show a strong image bias (69.57% and 74.98% respectively), while InternVL 1.5 and LLaVa-1.5 favor text-grounded responses (52.40% and 44.68%, respectively). This suggests different architectural or training approaches lead to systematic preferences for one modality over another when resolving conflicts.

---

[1]Table 6 in App. C reports results on a dataset containing both conflicting and non-conflicting samples.

Table 1: Percentage of predictions matching the respective answer for the multiple-choice question answering task. Last column denotes cases where no match with any of the answers was found. Top-tier closed-source models achieve $> 85\%$ conflict detection accuracy, while most open-source models fail, revealing systematic modality biases. The error bars show standard deviation.

| Model | Conflict ($\uparrow$) | Image | Text | Distractor | Incorrect |
|---|---|---|---|---|---|
| *Closed-source models* | | | | | |
| GPT 5 | $86.78_{\pm 0.89}$ | $1.09_{\pm 0.29}$ | $10.71_{\pm 0.86}$ | $0.15_{\pm 0.11}$ | $1.23_{\pm 0.31}$ |
| GPT 4.1 Mini | $13.56_{\pm 0.94}$ | $69.57_{\pm 1.30}$ | $12.26_{\pm 0.93}$ | $4.53_{\pm 0.58}$ | 0.00 |
| Gemini 2.5 Pro | $88.48_{\pm 0.89}$ | $2.87_{\pm 0.49}$ | $7.97_{\pm 0.75}$ | $0.39_{\pm 0.18}$ | $0.24_{\pm 0.13}$ |
| Gemini 2.5 Flash Lite | $7.51_{\pm 0.75}$ | $74.98_{\pm 1.22}$ | $15.57_{\pm 0.98}$ | $2.02_{\pm 0.40}$ | 0.00 |
| *Open-source models* | | | | | |
| InstructBlip-T5-xxl | $63.87_{\pm 1.21}$ | $12.72_{\pm 0.87}$ | $19.99_{\pm 1.01}$ | $3.35_{\pm 0.47}$ | $0.07_{\pm 0.07}$ |
| InternVL 1.5 | $16.71_{\pm 0.99}$ | $25.49_{\pm 1.12}$ | $52.40_{\pm 1.25}$ | $2.08_{\pm 0.36}$ | $3.27_{\pm 0.45}$ |
| MiniGPT4-7b | $2.20_{\pm 0.37}$ | $10.27_{\pm 0.80}$ | $29.79_{\pm 1.13}$ | $5.42_{\pm 0.59}$ | $52.30_{\pm 1.31}$ |
| LRV-MiniGPT4-7b | $2.15_{\pm 0.39}$ | $13.28_{\pm 0.88}$ | $26.35_{\pm 1.14}$ | $6.49_{\pm 0.67}$ | $51.74_{\pm 1.32}$ |
| Phi3-vision-128k | $1.06_{\pm 0.25}$ | $13.40_{\pm 0.86}$ | $25.19_{\pm 1.12}$ | $0.80_{\pm 0.23}$ | $59.63_{\pm 1.27}$ |
| Qwen2vl-7b | $1.21_{\pm 0.28}$ | $45.39_{\pm 1.30}$ | $41.58_{\pm 1.27}$ | $0.74_{\pm 0.22}$ | $11.15_{\pm 0.79}$ |
| LLaVa-1.5-7b | $0.13_{\pm 0.09}$ | $32.28_{\pm 1.24}$ | $44.68_{\pm 1.32}$ | $2.99_{\pm 0.45}$ | $19.99_{\pm 1.07}$ |

*(iii) Instruction following challenges.* Several open-ended models exhibit high "Incorrect" response rates (Phi3-vision-128k at 59.63%, MiniGPT4-7b at 52.30%), indicating difficulty in generating responses that match the required format. This suggests challenges in instruction following alongside the core reasoning task.

**Category-specific performance analysis.** To obtain a more fine-grained perspective of the performance differences, we analyze performance across object and attribute categories using four representative models. Fig. 5 presents the conflict detection accuracy for two closed-source (GPT-5, Gemini 2.5 Pro) and two open-source (InstructBLIP-T5-xxl, InternVL 1.5) models across different object and attribute categories (detailed results in Table 8 and 7 in App. C).

*Object category patterns.* Top–tier models perform strongest on Animals (GPT-5 with 97.32%, Gemini 2.5 Pro with 98.65%) and weakest on Household items ($\sim$78% for both). InstructBLIP-T5-xxl maintains consistent 60-77% performance across categories, while InternVL 1.5 shows uniformly poor results with slight advantages in Vehicles and Food.

*Attribute category patterns.* Environmental attributes prove most challenging for all models, while Colors are more easily detected. Numbers reveal an interesting capability difference between the leaders: Gemini 2.5 Pro substantially outperforming GPT-5 (85.83% vs 75.70%). InternVL 1.5 struggles across all attributes, while InstructBLIP-T5-xxl maintains moderate performance.

### 4.3 OPEN-ENDED QUESTION ANSWERING

We evaluate model performance on the open-ended question answering task, where models generate free-form responses to the same questions used in the multiple-choice evaluation.

**Evaluation protocol.** We use relaxed string matching (see App A.2) to classify responses into "Conflict", "Incorrect", image-grounded and text-grounded categories (see Table 2 in App C). LLM-as-a-judge evaluation with Gemini 2.5 Pro and GPT-5 confirms high agreement with string matching, validating the robustness of the evaluation (see Table 10 in App C).

**Performance evaluation.** Table 2 presents a summary of the conflict detection rates and incorrect response rates across several open– and closed-source models. Leading closed-source models maintain strong performance, with Gemini 2.5 Pro showing improved performance in the open-ended format (91.59% vs 88.45% in multiple-choice), while GPT-5 exhibits a slight decrease (81.16% vs 86.78%) primarily due to increased "Incorrect" (empty) responses. Notably, GPT-4.1 Mini shows improved performance (39.93%) compared to its multiple-choice results (13.56%).

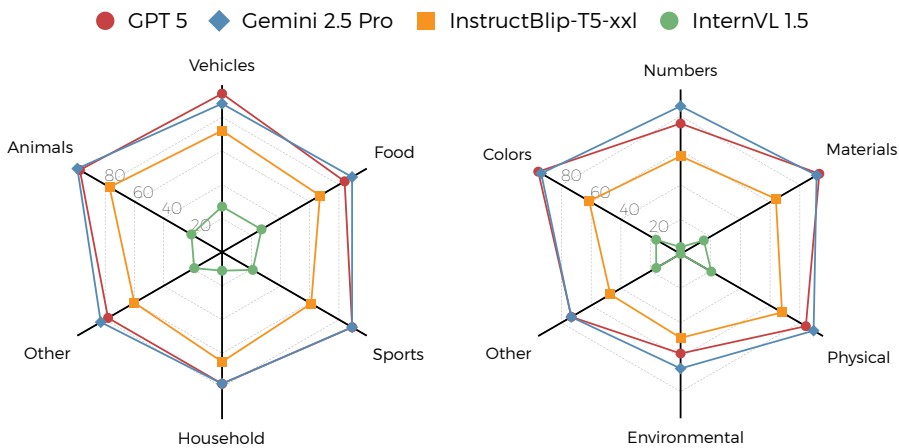

Figure 5: Category-specific performance on multiple-choice QA. We show contradiction detection accuracy across object categories (left) and attribute categories (right) for four representative models.

Open-source models demonstrate poor conflict detection capabilities, with most achieving near-zero performance, except InternVL 1.5 (18.76%). This stark performance gap suggests that current open-source models lack the fundamental reasoning capabilities required for cross-modal conflict detection. mPLUG-Owl-1 exhibits an exceptionally high incorrect response rate (61.61%), indicating severe instruction-following and response generation issues.

*Modality preference.* Fig. 6 shows the distribution of image– and text-grounded responses when models fail to detect contradictions. Several models demonstrate overwhelming text preference: mPLUG-Owl-2 shows extreme text bias (97.03%), while InternVL 1.5 and LLaVa-1.5-7b also lean heavily textual cues (55.51% and 62.10% respectively). In contrast, GPT-4.1 Mini and Gemini 2.5 Flash Lite shows moderate image preference (40.39% and 49.73%).

*Category-specific analysis.* Category-specific patterns in the open-ended task largely mirror those in multiple-choice evaluations (see Tables 11 and 12 in App. C).

Table 2: Conflict detection rate and rate of incorrect format outputs on the open ended task.

| Model | Conflict ($\uparrow$) | Incorrect ($\downarrow$) |
|---|---|---|
| GPT 5 | $81.16_{\pm 1.10}$ | $8.24_{\pm 0.74}$ |
| GPT 4.1 Mini | $39.93_{\pm 1.34}$ | $5.53_{\pm 0.65}$ |
| Gemini 2.5 Pro | $91.59_{\pm 0.77}$ | $0.31_{\pm 0.16}$ |
| Gemini 2.5 Flash Lite | $20.94_{\pm 1.14}$ | $9.32_{\pm 0.82}$ |
| InternVL 1.5 | $18.76_{\pm 1.10}$ | $6.33_{\pm 0.67}$ |
| mPLUG-Owl-1 | $6.56_{\pm 0.69}$ | $61.61_{\pm 1.35}$ |
| mPLUG-Owl-2 | $1.41_{\pm 0.33}$ | $0.94_{\pm 0.27}$ |
| LLaVa-1.5-7b | $0.00$ | $4.98_{\pm 0.59}$ |

Figure 6: Modality preference in open ended QA. Models show varying modality preferences when failing to detect contradictions.

## 4.4 LoRA Finetuning

The poor performance of open-ended models on CROSSCHECK motivates targeted fine-tuning as a potential remedy. To test this, we select the two worst-performing models, LLaVa-1.5-7b and mPLUG-Owl, and investigate whether direct exposure to conflict detection examples can improve their reasoning capabilities. Implementation details in App A.3.

**Training data.** We construct ~30k raw training samples following the pipeline described in § 3.2. After applying the quality control measures detailed in § 3.3, which remove ambiguous or unanswerable cases, we retain ~15k high-quality examples. To study data quality vs. quantity effects, we compare fine-tuning on the ~30k full set vs. the ~15k filtered subset.

**Evaluation metrics.** We evaluate models on: i) conflict (↑): contradiction detection rate on the CROSSCHECK test set; ii) no conflict (↑): performance on samples with consistent image-text pairs (using original captions) to measure false positive rates; iii) overall (↑): true positive rate across both contradiction and non-contradiction cases; and iv) incorrect (↓): frequency of invalid outputs on CROSSCHECK that fail to match with either contradiction or modality-specific answers.

**Results and analysis.** Table 3 shows that fine-tuning significantly boosts contradiction detection for both LLaVa-1.5-7b and mPLUG-Owl-1. For LLaVa, performance jumps from 0.00% to **76.86%** on contradiction samples with filtered training data, while maintaining strong accuracy on non-contradiction inputs (91.21%). Error rates also drop sharply (5.00% → 0.47%). Training on un-filtered ∼30k data yields lower contradiction accuracy (51.84%) and a higher error rate (8.21%), highlighting that data quality outweighs quantity. mPLUG-Owl-1 also improves substantially, from 6.56% to 57.45% contradiction accuracy, with error rates reduced from 61.63% to 7.09%. Inter-estingly, unfiltered training gives higher non-contradiction accuracy (75.12% vs 63.13%). Overall, these results confirm that targeted fine-tuning enables open-ended models to acquire robust multi-modal conflict detection capabilities.

Table 3: Fine-tuning results for multimodal conflict detection. We finetune LLaVa-1.5-7b and mPLUG-Owl-1 using ∼30k generated and ∼15k filtered samples. Columns report accuracy on conflicting and non-conflicting samples, overall accuracy across both, and the rate of unmatched outputs. Both models show substantial gains after fine-tuning.

| Model | Conflict (↑) | No conflict (↑) | Overall (↑) | Incorrect (↓) |
|---|---|---|---|---|
| LLaVa-1.5-7b | 0.00 | $92.83_{\pm 0.73}$ | $46.44_{\pm 0.97}$ | $5.00_{\pm 0.61}$ |
| LLaVa-1.5-7b-ft | $76.86_{\pm 1.16}$ | $91.21_{\pm 0.83}$ | $84.08_{\pm 0.74}$ | $0.47_{\pm 0.19}$ |
| LLaVa-1.5-7b-ft-30k | $51.84_{\pm 1.40}$ | $88.71_{\pm 0.87}$ | $70.27_{\pm 0.90}$ | $8.21_{\pm 0.77}$ |
| mPLUG-Owl-1 | $6.56_{\pm 0.70}$ | $31.62_{\pm 1.33}$ | $19.10_{\pm 0.78}$ | $61.63_{\pm 1.32}$ |
| mPLUG-Owl-1-ft | $57.45_{\pm 1.41}$ | $63.13_{\pm 1.34}$ | $60.28_{\pm 0.97}$ | $7.09_{\pm 0.72}$ |
| mPLUG-Owl-1-ft-30k | $50.61_{\pm 1.43}$ | $75.12_{\pm 1.22}$ | $62.85_{\pm 0.95}$ | $3.91_{\pm 0.53}$ |

## 5 CONCLUSION

We introduced CROSSCHECK, a diagnostic benchmark that systematically evaluates MM-LLM's ability to detect contradictions between visual and textual inputs. Unlike traditional datasets that treat a single modality – typically the image – as ground truth, CROSSCHECK requires models to critically evaluate both visual and textual information as equally valid sources, and detect inconsis-tencies that may cause hallucinations. Through our systematic construction pipeline, we generated controlled contradictions, paired with targeted evaluation questions and carefully validated answer sets. Our comprehensive experiments reveal a stark performance gap between leading closed-source models and open-source models, expose systematic modality biases across model architectures, and demonstrate that targeted fine-tuning with high-quality data can substantially enhance conflict de-tection capabilities.

**Limitations and future work.** While CROSSCHECK provides a controlled setting for studying multimodal inconsistencies, it has several limitations. First, it focuses on object– and attribute–level contradictions, excluding other inconsistencies, such as spatial relations, temporal sequences, ac-tions, events, or broader contextual contradictions. Second, building from everyday scenes in MS COCO may limit domain diversity; extending to specialized domains like medical, scientific, or technical content could improve generalization to professional applications. Finally, despite multi-stage filtering, some synthetic examples may contain lower quality outputs. Future work could inte-grate stronger validation mechanisms, including cross-model consistency checks or multi-annotator consensus, to enhance dataset reliability.

Despite these limitations, CROSSCHECK offers a valuable foundation for investigating fine-grained multimodal reasoning capabilities. We hope CROSSCHECK serves as a valueable tool for developing models that not only perceive multimodal inputs, but also critically reason about their consistency, paving the way for more trustworthy and reliable multimodal AI systems.

## ETHICS STATEMENT

This research focuses on evaluating and improving multimodal AI systems' ability to detect inconsistencies between visual and textual information—a capability critical for safe deployment in real-world applications. Our work aims to identify and address systematic limitations in current models that could lead to overconfident responses in the presence of conflicting information. The synthetic dataset construction process uses publicly available MS COCO images under their established usage terms and generates text modifications that do not introduce harmful content or biases. Human validation was conducted with clear guidelines, described in App G.

## REPRODUCIBILITY STATEMENT

We provide comprehensive implementation details including: i) complete dataset construction pipeline with prompts and quality control procedures; ii) detailed evaluation protocols for both multiple-choice and open-ended formats; iii) LoRA fine-tuning hyperparameters and training procedures; iv) statistical analysis methods including bootstrap resampling procedures. Code, data, and detailed experimental configurations will be made publicly available upon publication.

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

# Supplementary Material
# 📈✕📑 CROSSCHECK: A Vision-Language Conflict Detection Benchmark

The supplementary material is organized as follows:

- Implementation details (App. A)
- The prompts for data generation, quality check and model evaluation (App. B)
- Additional experiments and results on CROSSCHECK(App. C)
- Comprehensive list of the changed words (from original to conflictin caption) categorized into objects/attributes (App. D and App. E)
- Qualitative examples of CROSSCHECK question types (App. F)
- Details about the human validation of the test set (App. G)
- Details about LLM usage (App. H)
- Broader impact (App. I)

## A  IMPLEMENTATION DETAILS

### A.1  RELAXED STRING MATCHING FOR MULTIPLE CHOICE TASKS

Our evaluation protocol employs a hierarchical matching approach that accommodates various response formats. The matching process follows three sequential steps:

 (i) **Bracketed choices (strict matching)**: Responses like (A), (B), (C), (D) are matched directly to answer categories. If multiple bracketed options appear, the response is marked incorrect.
 (ii) **Single letters**: Responses like A–D (without brackets) are normalized and matched to the expected choices.
(iii) **Keyword matching**: If neither of the above applies, we check whether the response contains the exact answer text as a standalone word (case-insensitive, using word boundaries).

### A.2  RELAXED STRING MATCHING FOR OPEN ENDED TASKS

The open-ended question answering task presents models with conflicting image-text pairs and asks them to provide free-form responses explaining what they observe. We classify model responses into four mutually exclusive categories.

---

**Four response categories**

1. Conflict detection: responses that correctly identify the contradiction between image and text content.
2. Image-grounded: responses that describe or rely primarily on visual information.
3. Text-grounded: responses that align with or prioritize textual information.
4. Incorrect: responses that are unrelated to the content, incoherent, or fail to address the question meaningfully.

---

We implement a multi-step normalization pipeline to handle the natural variability in open-ended responses:

- Tokenization: extract word tokens and convert to lowercase.
- Articles removal: filter out articles (the, a, an).
- Number normalization: map numeric tokens to their word equivalents (e.g., "1" → "one").
- Word stripping: remove common linguistic variations using a predefined mapping (e.g., "wooden" → "wood", "brightly" → "bright").

- Stemming: apply Porter stemming to reduce words to their root forms, handling morphological variations.

For each response category, we perform substring matching on the normalized text:

- Contradiction detection: check for presence of stemmed versions of "conflict" or "contradict" using word boundary matching.
- Modality-specific responses: match against normalized versions of the expected image-only or text-only answers for each sample.
- Boundary matching: use regex word boundaries to ensure whole-word matches and avoid partial matches.

The automated evaluation system was manually validated on a subset of responses to ensure classification accuracy.

### A.3    FINE-TUNING WITH LORA

To avoid the strong bias of flagging conflicts, we pick "conflicting caption" or "original caption" with equal probability during fine-tuning. The LoRA fine-tuning is conducted for LLaVa-1.5-7b and mPLUG-Owl-1, both in 1 epoch. Following their official repository [2], we adopt bf16 and gradient checkpointing for efficiency. All experiments are conducted in a single A100-64GB. Check LoRA settings and training hyperparameters in Table 4.

Table 4: LoRA hyperparameters for multimodal conflict detection fine-tuning.

|  | LLaVa-1.5-7b | mPLUG-Owl-1 |
|---|---|---|
| LoRA-$r$ | 128 | 8 |
| LoRA-$\alpha$ | 256 | 32 |
| dropout | 0.05 | 0.05 |
| sequ. length | 2048 | 2048 |
| batch size | 16 | 4 |
| learning rate | 2e-5 | 2e-5 |
| scheduler | cosine | cosine |
| warmup | 0.03 ratio | 50 steps |

## B    PROMPTS

### B.1    DATA GENERATION

In this section, we present the prompts employed for data generation. These prompts were carefully designed to elicit high-quality outputs from the model while controlling for specific linguistic and visual attributes.

---

**Generate conflicting caption and question**

You are an expert image caption editor and question generator. Your task is to modify existing image captions and then create subtle questions based on your modifications. Given an original image caption, you need to perform the following four steps:
**Step 1: Create a conflicting caption.**

- Take the provided original caption.
- Identify *one* key element (either a specific object or an attribute of an object, like its color, number, shape, material, or texture).

---

[2]LLaVa `https://github.com/haotian-liu/LLaVA`, mPLUG-Owl-1 `https://github.com/X-PLUG/mPLUG-Owl`

- Change *only this one element* to create a subtle, but noticeable, conflict or discrepancy. The rest of the caption must remain identical to the original.

- Ensure the conflict is a plausible, though incorrect, alternative (e.g., "red car" to "blue car," not "red car" to "flying car").

- Do not change words that have binary states (e.g., man–woman, open–close, dark–light, indoor–outdoor).

When changing an attribute:

- Only change **objective attributes** such as **color, number, shape, material, or texture**.

- Do not change **subjective or ambiguous attributes** such as beautiful, small, large, big, medium, moderate, modern, young, old, fast, slow, elegant, scary, tall, short, etc.

**Step 2: Track the changed words.**

- Identify the exact word(s) that were changed from the original caption.

- Record both the original word(s) and the replacement word(s).

**Step 3: Identify the type of change.**

- Determine whether the change made in Step 1 was to an "object" (e.g., "cat" changed to "dog") or an "attribute" (e.g., "white" cat changed to "black" cat).

**Step 4: Generate a subtle question.**

- Based on your *newly created conflicting caption*, formulate a question.

- This question must subtly hint at the conflicting element without directly stating that something is wrong or different.

- The question should encourage the user to focus on the changed element.

Here is an example. Provide your responses in the exact JSON format shown:

```
User: "A fluffy white cat sitting on a red couch."
Model:
{
  "conflicting_caption": "A fluffy black cat sitting on a red couch.",
  "question": "What color is the cat sitting on a couch?",
  "change_type": "attribute",
  "changed_words": {
    "original": "white",
    "conflicting": "black"
  }
}
```

**Generate multiple choice answers**

Your task is to write **three answer choices** for a multiple-choice question that highlights a subtle conflict between two captions.

INSTRUCTIONS

Generate the following three answer options:

1. **image_only_answer**: The answer that fits the original caption.

2. **text_only_answer**: The answer that matches the conflicting caption.

3. **irrelevant_but_plausible**: A plausible distractor that doesn't appear in either the original or the conflicting caption, but is contextually reasonable.

   - Make sure the distractor is NOT ambiguous or vague (avoid words such as *several, afternoon, medium, moderate, thing, stuff*).

Ensure that all answers are:

- Concise (preferably 1–3 words),
- Mutually exclusive,
- Plausible in context, but not synonymous to each other.

Only output a JSON with the following fields:

```
{
  "image_only": "...",
  "text_only": "...",
  "irrelevant_but_plausible": "..."
}
```

## B.2 DATA FILTERING

This section describes the data filtering procedures applied to ensure the quality and consistency of the generated dataset. We perform multiple checks, including attribute verification, object validation, answer consistency, and question clarity. Each filtering step is designed to identify and remove entries that are ambiguous, irrelevant, or inconsistent, thereby maintaining the reliability of the dataset for downstream evaluation and analysis.

---

**Attribute check**

You are given two words: `original` and `conflicting`. Perform the following two steps:

1. Decide if each word is an **attribute** (a descriptive property, e.g., red, tall, beautiful) or **not an attribute** (an object, e.g., car, tree).
2. If both are attributes, classify them as:
   - **Objective** = measurable, factual, observable (e.g., red, square, wooden, three).
   - **Subjective** = opinion-based or interpretive (e.g., beautiful, small, moderate, large, big, medium, modern, young, old, fast, slow, elegant, scary, tall, short, stylish, fancy, cheap, impressive).

Only output a JSON with the following fields:

```
{
  "change_is_attribute": "Yes/No",
  "change_is_objective": "Yes/No"
}
```

---

**Object check**

You are given two words: `original` and `conflicting`.
Your task is to check the quality of the conflicting word in relation to the original:

1. **Object check:** Determine whether each word is an object (a tangible or identifiable thing/entity, e.g., car, apple, chair).
   - Are the two words objects?
2. **Synonymy check:** Is the conflicting object a synonym or near-synonym of the original?
3. **Ambiguity check:** Is the conflicting object ambiguous or vague (e.g., "thing", "object", "stuff")?
4. **Contextual relevance:** Does the conflicting object make sense in the same scene as the original?

Only output a JSON with the following fields:

```
{
```

---

```
    "change_is_object": "Yes/No",
    "change_is_synonym": "Yes/No",
    "change_is_ambiguous": "Yes/No",
    "change_is_relevant": "Yes/No"
}
```

## Answers check

You are given three words/phrases. For each word/phrase, check the following:

1. **Synonymy check:** Is one of the words/phrases a synonym or near-synonym of the other two?

2. **Ambiguity check:** Is any of the words/phrases ambiguous or vague (e.g., "several", "afternoon", "medium", "thing")?

3. **Contextual relevance:** Are all words/phrases contextually relevant and objective (not subjective or off-topic)?

4. **Visual check:** Can each word/phrase be directly observed in an image? Examples of visual words include:
   - Attributes of objects (number, color, shape, size, material)
   - Object categories (car, chair, dog)
   - Spatial relations (on top of, next to)
   - Scenes (beach, kitchen)

   Examples of non-visual words include:
   - Temporal concepts (afternoon, tomorrow)
   - Abstract states (freedom, happiness)
   - Non-observable attributes (brand, taste, temperature)
   - Subjective labels (beautiful, boring, large)

```
{
    "ans_is_synonym": "Yes/No",
    "ans_is_ambiguous": "Yes/No",
    "ans_is_relevant": "Yes/No",
    "ans_is_visible": "Yes/No"
}
```

## Questions check

You are given a question, a set of changed words, and three possible answers. Your task is to check the question based on three criteria:

1. **Ambiguity check:** Is the **question** clear, specific, and unambiguous?

2. **Focus check:** Does the question explicitly ask about the **changed words**? *Examples:*
   - Question: "What is the color of something?" Changed words: green, blue → Output: Yes
   - Question: "How many items are there?" Changed words: three, four → Output: Yes

3. **Answerability:** Are all three candidate **answers** semantically and contextually compatible with the question? *Examples:*
   - Question: "What is the color?" Answers: green, blue, red → Output: Yes
   - Question: "What is the gender?" Answers: man, woman, child → Output: No

Only output a JSON object with the following fields:

```
{
  "question_is_clear": "Yes/No",
  "question_is_focused": "Yes/No",
  "question_is_answerable": "Yes/No"
}
```

## B.3 MODEL EVALUATION

This section outlines the procedures used to evaluate model performance across different tasks. We describe the evaluation instructions for multiple-choice and open-ended questions, as well as the framework in which a large language model (LLM) acts as a judge to assess response quality.

---

**Multiple choice instruction**

You are given an image and a textual description of the image.
Use *both* the visual and textual information to answer the multiple-choice question below as accurately as possible.

- Choose the option that is best supported by **both sources**.
- If the image and the description contradict each other, or if one is missing key information, select the option "Conflicting information – cannot answer."
- Answer only with the letter corresponding to your choice in parentheses: (A), (B), (C), or (D).
- Do not include any other text.

---

**Open-ended instruction**

You are given an image and a textual description of the image.
Use *both* the visual and textual information to answer the question below as accurately as possible.

- Only answer if your response can be clearly supported by **both sources**.
- If the image and the description contradict each other, or if one is missing key information, output "Conflicting information – cannot answer"
- Otherwise, provide the answer (less than 15 words).

---

**LLM-as-a-judge**

You are an evaluator. Given two reference answers (image_only and text_only) and a model prediction, decide which category the prediction belongs to:

1. **IMAGE** — if the prediction semantically matches the image_only answer.
2. **TEXT** — if the prediction semantically matches the text_only answer.
3. **CONFLICT** — if the prediction explicitly refers to a contradiction, conflict, or states that both cannot be true.
4. **NONE** — if the prediction matches neither answer and does not indicate a conflict.

Ignore minor differences in phrasing, synonyms, plural/singular forms, or capitalization. Return only one label: **IMAGE**, **TEXT**, **CONFLICT**, or **NONE**.
**Examples:**

```
Image-only answer: polar bear
Text-only answer: brown bear
Prediction: Brown bear
Output: TEXT
```

```
Image-only answer: Black
Text-only answer: Blue
Prediction: Conflicting information { cannot answer.
Output: CONFLICT

Image-only answer: dog
Text-only answer: cat
Prediction: dog
Output: IMAGE

Image-only answer: red
Text-only answer: green
Prediction: yellow
Output: NONE
```

# C  EXPERIMENTS

In this section we report additional experiments and detailed analysis of model performance on CROSSCHECK. We provide comprehensive results across both multiple-choice and open-ended evaluation formats, including category-specific breakdowns and validation of our evaluation methodology through LLM-as-a-judge assessment.

## C.1  MULTIPLE CHOICE QUESTION ANSWERING

**Performance evaluation.** Table 5 shows percentage of predictions matching the respective answer for the multiple-choice question answering task using relaxed string matching for evaluation. This format tests models' ability to recognize conflicts when provided with explicit options, including the correct "Conflicting information – cannot answer" choice.

Figure 7 depicts the modality preference of various models, revealing systematic biases toward either visual or textual information. Leading closed-source models (GPT-5, Gemini 2.5 Pro) show minimal bias, while their lighter variants (GPT-4.1 Mini, Gemini Flash Lite) demonstrate strong image preference. Open-source models show varying degrees of modality preference, with some strongly favoring text (InterVL-1.5, LLaVa-1.5-7b) and others showing more balanced distributions.

Table 5: Percentage of predictions matching the respective answer for the multiple-choice question answering task using relaxed string matching. Last column denotes cases where no match with any of the answers was found.

| Model | Conflict ($\uparrow$) | Image ($\downarrow$) | Text ($\downarrow$) | Distractor ($\downarrow$) | Incorrect ($\downarrow$) |
|---|---|---|---|---|---|
| Phi3-vision-128k | $1.29_{\pm0.28}$ | $27.51_{\pm1.10}$ | $53.65_{\pm1.35}$ | $1.19_{\pm0.28}$ | $16.30_{\pm0.96}$ |
| MiniGPT4-7b | $2.18_{\pm0.38}$ | $23.50_{\pm1.13}$ | $51.11_{\pm1.27}$ | $9.88_{\pm0.75}$ | $20.50_{\pm1.04}$ |
| mPLUG-Owl-2 | $0.00$ | $0.72_{\pm0.22}$ | $77.92_{\pm1.06}$ | $0.20_{\pm0.12}$ | $21.10_{\pm1.05}$ |
| LRV-MiniGPT4-7b | $2.16_{\pm0.39}$ | $24.10_{\pm1.06}$ | $42.77_{\pm1.24}$ | $10.69_{\pm0.78}$ | $30.41_{\pm1.21}$ |
| InternVL 1.5 | $17.05_{\pm0.99}$ | $26.35_{\pm1.12}$ | $53.65_{\pm1.31}$ | $2.34_{\pm0.38}$ | $0.80_{\pm0.23}$ |
| InstructBlip-T5xxl | $63.86_{\pm1.23}$ | $12.67_{\pm0.84}$ | $19.89_{\pm1.02}$ | $3.30_{\pm0.47}$ | $0.06_{\pm0.06}$ |
| LLaVa-1.5-7b | $0.13_{\pm0.09}$ | $38.98_{\pm1.23}$ | $57.89_{\pm1.29}$ | $3.09_{\pm0.43}$ | $0.00$ |
| Qwen2vl-7b | $1.27_{\pm0.29}$ | $50.12_{\pm1.31}$ | $47.64_{\pm1.27}$ | $1.01_{\pm0.25}$ | $0.00$ |

**Evaluation with non-conflicting samples.** To better understand the robustness of MM-LLMs in detecting multimodal contradictions, we evaluate models on data that includes both conflicting and non-conflicting samples. This experiment uses the same models and evaluation protocols but incorporates samples where the original caption (without modifications) is presented alongside the image, creating scenarios where no contradiction exists. In these non-conflicting cases, models should respond with the image–grounded answer.

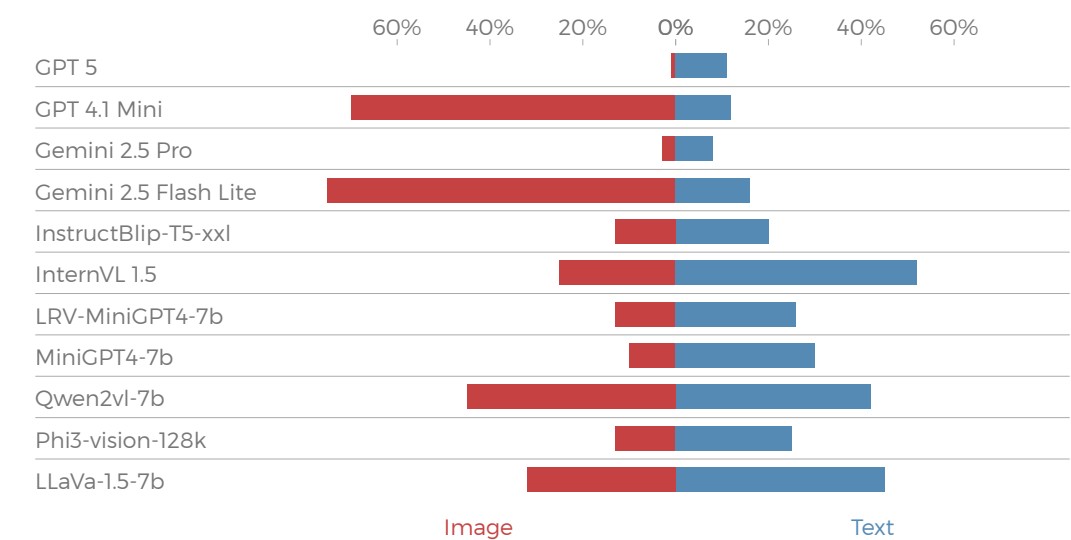

Figure 7: Modality preference patterns in multiple-choice QA. Most models exhibit systematic biases, favoring either visual (red) or textual (blue) information when faced with contradictions.

Table 6: Performance of various models on multiple-choice QA including conflicting and non-conflicting samples. *Conflict* shows accuracy on contradictory samples, *No conflict* on consistent samples, and *Overall* is the true positive rate across both.

| Model | Conflict ($\uparrow$) | No conflict ($\uparrow$) | Overall ($\uparrow$) |
|---|---|---|---|
| ***Strict string matching*** | | | |
| InstructBlip-T5xxl | $63.54_{\pm1.36}$ | $42.66_{\pm1.34}$ | $53.18_{\pm0.98}$ |
| InternVL1.5 | $16.25_{\pm1.02}$ | $92.51_{\pm0.73}$ | $54.27_{\pm0.95}$ |
| MiniGPT4-7b | $2.32_{\pm0.42}$ | $35.39_{\pm1.33}$ | $18.80_{\pm0.78}$ |
| MiniGPT4-7b-LRV | $2.18_{\pm0.42}$ | $36.01_{\pm1.32}$ | $19.04_{\pm0.77}$ |
| Phi3-vision-128k | $0.94_{\pm0.27}$ | $48.77_{\pm1.40}$ | $24.80_{\pm0.86}$ |
| Qwen2vl-7b | $1.25_{\pm0.30}$ | $94.55_{\pm0.65}$ | $47.82_{\pm0.98}$ |
| LLaVa1.5-7b | $0.07_{\pm0.08}$ | $63.84_{\pm1.36}$ | $31.95_{\pm0.92}$ |
| ***Relaxed string matching*** | | | |
| InstructBlip-T5xxl | $63.59_{\pm1.35}$ | $42.70_{\pm1.42}$ | $53.17_{\pm0.98}$ |
| InternVL1.5 | $16.71_{\pm0.99}$ | $92.98_{\pm0.70}$ | $54.71_{\pm1.01}$ |
| MiniGPT4-7b | $2.34_{\pm0.42}$ | $63.13_{\pm1.33}$ | $32.62_{\pm0.96}$ |
| MiniGPT4-7b-LRV | $2.17_{\pm0.41}$ | $55.71_{\pm1.36}$ | $28.89_{\pm0.87}$ |
| Phi3-vision-128k | $1.08_{\pm0.30}$ | $89.45_{\pm0.83}$ | $45.13_{\pm1.02}$ |
| Qwen2vl-7b | $1.31_{\pm0.31}$ | $97.02_{\pm0.47}$ | $49.04_{\pm0.96}$ |
| LLaVa1.5-7b | $0.08_{\pm0.08}$ | $90.44_{\pm0.81}$ | $45.24_{\pm0.99}$ |

We report the results of this experiment in Table 6, both with strict and relaxed string matching. The "Conflict" column shows accuracy on samples with visual-textual contradictions, "No conflict" column shows accuracy on samples with consistent information, and "Overall" represents the true positive rate across both conflicting and non-conflicting samples. InternVL1.5, despite low conflict detection (16%), achieves exceptional performance on non-conflicting samples (93%), suggesting the model model can appropriately respond to non-conflicting information but fails at conflict identification. Conversely, InstructBlip-T5xxl shows stronger conflict detection (64%) but weaker non-conflict performance (43%). LLaVa-1.5-7b achieves near-zero conflict detection but achieves moderate performance on non-conflicting samples (63.84%).

Comparing the strict vs. relaxed string matching evaluation reveals that most models demonstrate minimal performance changes. However, several models show substantial improvements in non-

conflict performance under relaxed matching. Models like LLaVa1.5-7b and Phi3-vision-128k appear to understand task requirements but struggle with strict answer formatting, leading to substantial underestimation of their capabilities under strict evaluation.

**Category-specific performance analysis.** To understand how different types of contradictions affect performance, we analyze results across semantic categories. Tables 8 and 7 show an overview of the performance split across the object and attribute categories, respectively. This breakdown reveals whether models struggle more with certain types of contradictions (e.g., color vs. environmental characteristics) and helps identify systematic weaknesses in multimodal reasoning capabilities.

Table 7: Object category performance breakdown for multiple choice QA. We report mean $\pm$ standard deviation. Results across five major categories reveal category-specific strengths and weaknesses in multimodal conflict detection. Performance variations suggest that different object types pose varying difficulty levels for conflict detection, potentially due to visual saliency, semantic complexity, or training data distribution.

| Model | Animals | Vehicles | Food | Sports | Household | Other |
|---|---|---|---|---|---|---|
| GPT 5 | $97.32_{\pm1.33}$ | $93.91_{\pm2.97}$ | $83.88_{\pm3.92}$ | $89.07_{\pm3.15}$ | $78.38_{\pm3.25}$ | $77.58_{\pm4.15}$ |
| GPT 4.1 Mini | $3.47_{\pm1.49}$ | $10.38_{\pm3.75}$ | $14.89_{\pm3.72}$ | $1.06_{\pm1.07}$ | $8.54_{\pm2.39}$ | $8.80_{\pm2.75}$ |
| Gemini 2.5 Pro | $98.65_{\pm0.90}$ | $88.06_{\pm4.03}$ | $88.56_{\pm3.46}$ | $89.12_{\pm3.22}$ | $77.74_{\pm3.60}$ | $82.62_{\pm3.66}$ |
| Gemini 2.5 Flash Lite | $8.25_{\pm2.28}$ | $6.12_{\pm2.90}$ | $10.36_{\pm3.38}$ | $3.37_{\pm1.91}$ | $10.58_{\pm2.60}$ | $4.84_{\pm2.10}$ |
| InstructBlip-T5xxl | $76.66_{\pm3.36}$ | $71.70_{\pm5.52}$ | $66.60_{\pm4.98}$ | $60.82_{\pm5.11}$ | $64.52_{\pm3.85}$ | $59.88_{\pm4.83}$ |
| InternVL1.5 | $21.16_{\pm3.31}$ | $27.00_{\pm5.59}$ | $27.70_{\pm4.77}$ | $20.91_{\pm4.31}$ | $11.17_{\pm2.56}$ | $18.95_{\pm3.97}$ |
| Phi3-vision-128k | $2.05_{\pm1.17}$ | $0.00$ | $0.00$ | $1.17_{\pm1.14}$ | $0.70_{\pm0.73}$ | $2.05_{\pm1.36}$ |
| MiniGPT4-7b | $1.35_{\pm0.96}$ | $2.95_{\pm2.10}$ | $3.41_{\pm1.93}$ | $3.43_{\pm1.89}$ | $2.05_{\pm1.20}$ | $1.94_{\pm1.36}$ |
| mPLUG-Owl-2 | $0.00$ | $0.00$ | $0.00$ | $0.00$ | $0.00$ | $0.00$ |
| LRV-MiniGPT4-7b | $2.81_{\pm1.43}$ | $1.49_{\pm1.47}$ | $1.15_{\pm1.17}$ | $1.10_{\pm1.13}$ | $1.41_{\pm1.00}$ | $2.01_{\pm1.40}$ |
| InternVL1.5 | $21.16_{\pm3.31}$ | $27.00_{\pm5.59}$ | $27.70_{\pm4.77}$ | $20.91_{\pm4.31}$ | $11.17_{\pm2.56}$ | $18.95_{\pm3.97}$ |
| LLaVa-1.5-7b | $0.00$ | $0.00$ | $0.00$ | $0.00$ | $0.00$ | $0.00$ |
| Qwen2vl-7b | $1.40_{\pm1.00}$ | $1.52_{\pm1.51}$ | $0.00$ | $0.00$ | $2.10_{\pm1.21}$ | $3.84_{\pm1.97}$ |

Table 8: Attribute category performance breakdown for multiple choice QA. We report mean $\pm$ standard deviation. Results demonstrate how models handle different descriptive properties, from concrete visual attributes (colors, materials) to more abstract characteristics (environmental conditions, physical properties). Notable performance gaps emerge between attribute categories, with colors generally being easier to detect than environmental descriptors.

| Model | Colors | Numbers | Materials | Physical | Environmental | Other |
|---|---|---|---|---|---|---|
| GPT 5 | $95.49_{\pm1.24}$ | $75.70_{\pm2.95}$ | $93.06_{\pm3.38}$ | $84.04_{\pm8.09}$ | $57.88_{\pm14.19}$ | $73.22_{\pm8.41}$ |
| GPT 4.1 Mini | $31.59_{\pm2.62}$ | $6.87_{\pm1.75}$ | $11.82_{\pm4.18}$ | $5.06_{\pm5.31}$ | $16.74_{\pm10.83}$ | $10.12_{\pm5.49}$ |
| Gemini 2.5 Pro | $93.50_{\pm1.37}$ | $85.83_{\pm2.45}$ | $91.24_{\pm3.67}$ | $89.21_{\pm7.22}$ | $66.58_{\pm13.63}$ | $73.34_{\pm7.89}$ |
| Gemini 2.5 Flash Lite | $9.42_{\pm1.66}$ | $4.42_{\pm1.47}$ | $6.78_{\pm3.24}$ | $20.96_{\pm9.44}$ | $8.63_{\pm7.99}$ | $10.15_{\pm5.57}$ |
| InstructBlip-T5xxl | $61.35_{\pm2.84}$ | $56.73_{\pm3.46}$ | $63.93_{\pm6.32}$ | $68.03_{\pm10.83}$ | $48.87_{\pm14.95}$ | $47.11_{\pm8.87}$ |
| InternVL1.5 | $16.40_{\pm2.13}$ | $3.94_{\pm1.44}$ | $15.60_{\pm4.73}$ | $20.59_{\pm9.24}$ | $0.00$ | $16.35_{\pm7.05}$ |
| Phi3-vision-128k | $1.57_{\pm0.72}$ | $0.00$ | $0.00$ | $0.00$ | $0.00$ | $3.31_{\pm3.29}$ |
| MiniGPT4-7b | $2.58_{\pm0.89}$ | $2.42_{\pm1.04}$ | $1.76_{\pm1.71}$ | $0.00$ | $0.00$ | $3.28_{\pm3.27}$ |
| mPLUG-Owl-2 | $0.00$ | $0.00$ | $0.00$ | $0.00$ | $0.00$ | $0.00$ |
| LRV-MiniGPT4-7b | $2.27_{\pm0.87}$ | $2.96_{\pm1.16}$ | $1.66_{\pm1.72}$ | $0.00$ | $0.00$ | $6.59_{\pm4.36}$ |
| LLaVa-1.5-7b | $0.00$ | $0.00$ | $0.00$ | $0.00$ | $0.00$ | $0.00$ |
| Qwen2vl-7b | $0.97_{\pm0.55}$ | $0.48_{\pm0.49}$ | $0.00$ | $0.00$ | $0.00$ | $0.00$ |

## C.2 OPEN ENDED QUESTION ANSWERING

Unlike multiple-choice tasks where models select from given options, open-ended questions require models to formulate their own responses. This format more closely mirrors real-world deployment scenarios where models must generate explanations or decisions without explicit guidance about potential conflicts.

We evaluate whether models can naturally identify contradictions between visual and textual inputs, or whether they default to following one modality while ignoring the other. The evaluation protocol categorizes responses into four classes: correctly identifying **conflicts**, following **image**-based information, adhering to **text**-based descriptions, or producing **incorrect/**unintelligible responses.

**Overall performance on open ended tasks.** This section presents a comprehensive analysis of open-ended conflict detection performance, summarizing key results from the main paper. Table 9 provides the complete breakdown of response categories, while Table 2 and Figure 6 in the main text focus on conflict detection rates and modality bias patterns separately.

Table 9: Evaluation of models on open ended QA using relaxed string matching. Columns indicate the percentage of predictions matching Conflict, Image or Text, with Incorrect denoting responses that match none of them.

| Model | Conflict ($\uparrow$) | Image ($\downarrow$) | Text ($\downarrow$) | Incorrect ($\downarrow$) |
|---|---|---|---|---|
| GPT 5 | $81.21_{\pm 1.08}$ | $1.11_{\pm 0.30}$ | $9.49_{\pm 0.81}$ | $8.21_{\pm 0.78}$ |
| GPT 4.1 Mini | $40.03_{\pm 1.37}$ | $40.39_{\pm 1.35}$ | $19.59_{\pm 1.09}$ | $5.53_{\pm 0.61}$ |
| Gemini 2.5 Pro | $91.61_{\pm 0.76}$ | $0.84_{\pm 0.25}$ | $7.78_{\pm 0.70}$ | $0.30_{\pm 0.15}$ |
| Gemini 2.5 Flash Lite | $20.94_{\pm 1.12}$ | $49.73_{\pm 1.35}$ | $21.64_{\pm 1.18}$ | $9.39_{\pm 0.82}$ |
| InternVL1.5 | $18.72_{\pm 1.11}$ | $22.67_{\pm 1.19}$ | $55.51_{\pm 1.42}$ | $6.31_{\pm 0.70}$ |
| mPLUG-Owl-1 | $6.57_{\pm 0.69}$ | $15.91_{\pm 1.03}$ | $24.17_{\pm 1.22}$ | $61.50_{\pm 1.38}$ |
| mPLUG-Owl-2 | $1.40_{\pm 0.33}$ | $1.02_{\pm 0.29}$ | $97.03_{\pm 0.48}$ | $0.93_{\pm 0.26}$ |
| LLaVa-1.5-7b | $0.00$ | $34.67_{\pm 1.29}$ | $62.10_{\pm 1.39}$ | $4.97_{\pm 0.63}$ |

**LLM-as-judge.** To assess the consistency and reliability of our relaxed string matching procedure for evaluating the open ended task, we ran an LLM-as-a-judge using two models: Gemini 2.5 Pro and GPT-5. For each sample, we determined whether each model's prediction matched one of the four categories: CONFLICT, IMAGE, TEXT, or NONE. We then recorded:

1. **Gemini 2.5 Pro "Yes"** – number of samples where Gemini assigned the category.

2. **GPT-5 "Yes"** – number of samples where GPT-5 assigned the category.

3. **At least one "Yes"** – number of samples where either model assigned the category.

4. **Both "Yes"** – number of samples where both models agreed on the category.

This setup allows quantifying both individual model performance and inter-model agreement. Results are presented in Table 10. The results reveal several trends, discussed below.

*High consistency across evaluation methods.* The comparison between the string matching evaluation in Table 9 and LLM-as-judge evaluation in Table 10 revels remarkably consistent results, demonstrating the reliability of both approaches. Across all models, the differences between string matching and LLM-based evaluation are minimal, typically within 1-3 percentage points. For instance, InternVL 1.5's conflict detection accuracy shows only a 1.79 point difference (18.72% string matching vs. 20.51% Gemini), while modality bias patterns remain nearly identical.

*Inter-judge agreement and reliability.* The strong agreement between Gemini 2.5 Pro and GPT-5 as judges (differences $< 1\%$ across most metrics) validates the robustness of LLM-based evaluation. This consistency suggests that both judge models apply similar semantic understanding when categorizing responses, reducing concerns about judge-specific biases.

These findings strengthen confidence in our evaluation methodology and suggest that either approach can reliably assess model performance on CROSSCHECK.

**Performance across object and attribute categories on open ended tasks.** We further analyze category-specific performance in the open-ended setting to understand how different types of contradictions affect free-form reasoning capabilities. This breakdown across object and attribute categories reveals whether the patterns observed in multiple-choice evaluation persist when models

Table 10: LLM-as-judge evaluation of predictions from four evaluated open-ended models, using Gemini 2.5 Pro and GPT-5. Results show high consistency between judges and close alignment with string matching results (Table 9).

| Evaluated Model | Judge | Conflict ($\uparrow$) | Image ($\downarrow$) | Text ($\downarrow$) | Incorrect ($\downarrow$) |
|---|---|---|---|---|---|
| InternVL1.5 | Gemini | $20.51_{\pm 1.08}$ | $23.83_{\pm 1.15}$ | $52.24_{\pm 1.37}$ | $3.43_{\pm 0.49}$ |
| | GPT-5 | $20.54_{\pm 1.14}$ | $23.20_{\pm 1.21}$ | $52.04_{\pm 1.38}$ | $4.23_{\pm 0.56}$ |
| | $\geq 1$ | $21.33_{\pm 1.09}$ | $24.29_{\pm 1.18}$ | $52.30_{\pm 1.39}$ | $4.43_{\pm 0.58}$ |
| | Both | $19.75_{\pm 1.14}$ | $22.81_{\pm 1.20}$ | $51.94_{\pm 1.42}$ | $3.19_{\pm 0.49}$ |
| mPLUG-Owl-1 | Gemini | $2.67_{\pm 0.45}$ | $14.24_{\pm 0.99}$ | $19.70_{\pm 1.11}$ | $61.85_{\pm 1.35}$ |
| | GPT-5 | $2.43_{\pm 0.43}$ | $13.49_{\pm 0.94}$ | $19.23_{\pm 1.07}$ | $64.89_{\pm 1.29}$ |
| | $\geq 1$ | $3.79_{\pm 0.53}$ | $14.75_{\pm 0.99}$ | $20.51_{\pm 1.14}$ | $65.95_{\pm 1.40}$ |
| | Both | $1.25_{\pm 0.31}$ | $12.97_{\pm 0.95}$ | $18.46_{\pm 1.11}$ | $60.79_{\pm 1.38}$ |
| mPLUG-Owl-2 | Gemini | $1.40_{\pm 0.33}$ | $0.78_{\pm 0.25}$ | $96.69_{\pm 0.50}$ | $1.08_{\pm 0.29}$ |
| | GPT-5 | $1.41_{\pm 0.34}$ | $0.71_{\pm 0.24}$ | $96.79_{\pm 0.47}$ | $1.10_{\pm 0.30}$ |
| | $\geq 1$ | $1.38_{\pm 0.34}$ | $0.77_{\pm 0.24}$ | $97.04_{\pm 0.48}$ | $1.33_{\pm 0.32}$ |
| | Both | $1.41_{\pm 0.33}$ | $0.71_{\pm 0.23}$ | $96.49_{\pm 0.50}$ | $0.85_{\pm 0.25}$ |
| LLaVa-1.5-7b | Gemini | $0.24_{\pm 0.13}$ | $35.89_{\pm 1.32}$ | $60.31_{\pm 1.36}$ | $3.56_{\pm 0.51}$ |
| | GPT-5 | $0.16_{\pm 0.11}$ | $36.11_{\pm 1.36}$ | $60.95_{\pm 1.34}$ | $2.70_{\pm 0.45}$ |
| | $\geq 1$ | $0.31_{\pm 0.16}$ | $36.22_{\pm 1.38}$ | $61.17_{\pm 1.36}$ | $3.66_{\pm 0.53}$ |
| | Both | $0.08_{\pm 0.08}$ | $35.76_{\pm 1.35}$ | $60.20_{\pm 1.32}$ | $2.64_{\pm 0.46}$ |

must generate their own explanations rather than select from predefined options. Tables 11 and 12 performance of several open source and closed source models on the open-ended task across object and attribute categories. The evaluation is performed using relaxed string matching.

Table 11: Object category performance of several open source and closed source models on the open-ended task using relaxed string matching evaluation. We report mean $\pm$ standard deviation.

| Model | Animals | Vehicles | Food | Sports | Household | Other |
|---|---|---|---|---|---|---|
| GPT 5 | $96.55_{\pm 1.50}$ | $89.56_{\pm 3.74}$ | $74.73_{\pm 4.55}$ | $86.82_{\pm 3.47}$ | $64.48_{\pm 4.12}$ | $67.53_{\pm 4.42}$ |
| GPT 4.1 Mini | $54.82_{\pm 4.21}$ | $43.27_{\pm 6.16}$ | $38.86_{\pm 5.29}$ | $38.13_{\pm 4.92}$ | $33.52_{\pm 3.98}$ | $31.38_{\pm 4.53}$ |
| Gemini 2.5 Pro | $98.67_{\pm 0.93}$ | $95.58_{\pm 2.39}$ | $92.99_{\pm 2.68}$ | $95.56_{\pm 2.12}$ | $84.72_{\pm 3.00}$ | $82.34_{\pm 3.82}$ |
| Gemini 2.5 Flash Lite | $21.24_{\pm 3.41}$ | $13.46_{\pm 4.17}$ | $11.34_{\pm 3.33}$ | $8.67_{\pm 2.93}$ | $16.66_{\pm 3.04}$ | $20.63_{\pm 4.08}$ |
| InternVL1.5 | $26.10_{\pm 3.63}$ | $31.37_{\pm 5.59}$ | $21.71_{\pm 4.17}$ | $31.54_{\pm 4.55}$ | $14.64_{\pm 2.93}$ | $20.56_{\pm 3.96}$ |
| mPLUG-Owl-1 | $4.77_{\pm 1.77}$ | $7.36_{\pm 3.01}$ | $10.31_{\pm 3.22}$ | $3.27_{\pm 1.83}$ | $5.55_{\pm 1.94}$ | $4.91_{\pm 2.19}$ |
| mPLUG-Owl-2 | $0.00$ | $0.00$ | $2.38_{\pm 1.63}$ | $1.02_{\pm 1.04}$ | $0.70_{\pm 0.69}$ | $2.91_{\pm 1.69}$ |
| LLaVa-1.5-7b | $0.00$ | $0.00$ | $0.00$ | $0.00$ | $0.00$ | $0.00$ |

Table 12: Attribute category performance of several open source and closed source models on the open-ended task using relaxed string matching evaluation. We report mean $\pm$ standard deviation.

| Model | Colors | Numbers | Materials | Physical | Environmental | Other |
|---|---|---|---|---|---|---|
| GPT 5 | $93.52_{\pm 1.41}$ | $67.33_{\pm 3.39}$ | $91.24_{\pm 3.81}$ | $68.17_{\pm 10.46}$ | $49.61_{\pm 12.07}$ | $76.63_{\pm 8.31}$ |
| GPT 4.1 Mini | $66.18_{\pm 2.75}$ | $3.36_{\pm 1.19}$ | $27.33_{\pm 6.00}$ | $5.45_{\pm 5.21}$ | $31.73_{\pm 11.54}$ | $50.01_{\pm 9.71}$ |
| Gemini 2.5 Pro | $96.48_{\pm 1.08}$ | $86.50_{\pm 2.34}$ | $96.60_{\pm 2.37}$ | $89.16_{\pm 7.11}$ | $62.86_{\pm 12.15}$ | $80.85_{\pm 7.62}$ |
| Gemini 2.5 Flash Lite | $34.21_{\pm 2.65}$ | $16.10_{\pm 2.58}$ | $22.24_{\pm 5.63}$ | $16.05_{\pm 8.49}$ | $12.22_{\pm 8.40}$ | $26.83_{\pm 8.70}$ |
| InternVL1.5 | $16.80_{\pm 2.06}$ | $13.15_{\pm 2.39}$ | $10.32_{\pm 3.87}$ | $0.00$ | $18.91_{\pm 9.80}$ | $7.74_{\pm 5.07}$ |
| mPLUG-Owl-1 | $7.79_{\pm 1.47}$ | $5.81_{\pm 1.60}$ | $6.87_{\pm 3.29}$ | $15.73_{\pm 8.64}$ | $0.00$ | $3.87_{\pm 3.87}$ |
| mPLUG-Owl-2 | $0.31_{\pm 0.31}$ | $1.46_{\pm 0.85}$ | $10.09_{\pm 3.89}$ | $5.17_{\pm 5.13}$ | $0.00$ | $0.00$ |
| LLaVa-1.5-7b | $0.00$ | $0.00$ | $0.00$ | $0.00$ | $0.00$ | $0.00$ |

# D  OBJECT CATEGORIES

This section presents the object categories used in our evaluation framework. The categories are organized into four main domains: animals, transportation, food, sports, and household items.

## D.1  ANIMALS

- **Domestic/Farm Animals:** dog, cat, sheep, cow, cows, horse, horses, pig, goat, goats, cattle, donkeys, chicken, bull

- **Wild Animals:** elephant, zebra, zebras, giraffe, giraffes, bear, rhino, rhinoceros, rhinoceroses, rhinos, bird, birds, monkey, camel, antelope, deer, bison, wildebeest, hippopotamus, hippos, polar bear, brown bear, parrot, owl, crow, pigeon, butterfly, octopus, shark, fish, worm

- **Multiple/General:** elephants, dogs, cats, ducks, animals, kitten, puppy

## D.2  TRANSPORTATION

- **Land Vehicles:** skateboard, skate board, bicycle, car, bus, scooter, motorcycle, train, truck, cars, tractor, automobile, motorcycles, bicycles, scooters, fire truck, police car, snowmobile, tow truck, pickup truck, train car, tour buses, bullet train

- **Air Vehicles:** plane, airplane, air plane, helicopter, fighter jet, commercial plane, fighter jets, commercial jets

- **Water Vehicles:** boat, boats, surf board, surfboard, surf boards, kayak, wakeboard

- **Transportation-Related:** bike, bikes, commercial jet, engine, trunk, trucks, road, tracks, track, windsail, parking meter

## D.3  FOOD

- **Fruits:** apples, bananas, banana, fruits, fruit, oranges, apple, pear, strawberries, blueberries, strawberry, banana peel, apple core

- **Vegetables:** vegetables, carrots, potatoes, broccoli, olives, tomatoes, carrot, cauliflower, onion rings, mushrooms, peppers, peas, spinach

- **Prepared Food:** pizza, burger, burgers, cake, hot dog, hot dogs, pastry, sandwich, donuts, cookies, food, noodles, hamburgers, hotdogs, cheese, sauce, sandwiches, quiche, meat, mead, beef, eggs, pasta, french fries, bread, hamburger, rice, cheeses, meats, fries, rice cake, cookie, pickle, piece of cake, slice of pizza, sausage

- **Drinks:** wine, beer, coffee, wine bottle, beer bottle, coffees, drinks, milk

- **Food Descriptors/Toppings:** toppings, sauces, greens, pepperoni

- **Food-Related Items:** bar b que, blender, bottle, bottles, bowls, chicken, dishes, fork, knife, olive, oysters, pears, pie, spoon, water, snails

## D.4  SPORTS

- **Sports Equipment:** ball, frisbee, snowboard, snow board, skis, ski, snowboards, baseball bat, baseball, tennis racket, tennis racquet, basketball, kite, kites, tennis ball, football, glove, bat, tennis, surfboard, surf board, surf boards, soccer, soccer ball, soccer balll, golf club, golf, golf ball, racket, racquet, shuttlecock, a snowboard, skateboard, skate board, bicycle

- **Sports Participants:** snowboarder, skier, skiier, batter, catcher, snowboarders, skiers, skateboarder, cyclist, surfer, kayaker, tennis players, basketball players, tennis player, basketball player, baseball player, football player, baseball players, football players, umpire, skateboarders, cyclists

- **Sports Venues:** tennis court, basketball court, skate park, ski lift

- **Sports-Related:** base ball, cricket, pitch, pitcher, ski lift, skis, surfer, track, wakeboard

## D.5 HOUSEHOLD ITEMS

- **Furniture:** chair, table, bed, sofa, couch, bench, coffee table, dining table, computer desk, nightstand, shelf, counter
- **Room Identifiers:** kitchen, bathroom, bedroom, living room, dining room
- **Bathroom Items:** toilet, sink, bathtub, shower, toothbrush, tooth brush, toilet tissue, soap, mirror, toilet bowl, shower curtain, towel rack, handicap bar
- **Kitchen Items:** fork, bowl, bowls, spoon, plate, plates, knife, knif, cup, trays, pots
- **Storage/Containers:** bag, trash can, laundry basket, baskets, mason jar, coffee cup, bottle, bottles, vases
- **Technology/Electronics:** phone, cell phone, laptop, laptops, keyboard, mouse, tablet, tablets, television, camera, phones, cellphones, wii, wii console, playstation, playstation console, xbox, remote, game remote, controller, refrigerator, oven, stove, microwave, dishwasher, washer, dryer, clothes washer, clothes dryer, blender, ice machine, coffee machine, printer, monitors, screen, clock, bell, ipod, microphones, speakers, equipment
- **Decor/Furnishing:** lamp, paintings, painting, picture frame, pillow, carpet, rug, sculptures, sculpture, statues, statutes, crosses, flags, flag
- **General Household:** furniture, window, door, towels, dishes, appliances, comb, rope, chain, scarf, mask, ties, scarves, backpack, coat, shirt, belt, hairbrush, aluminum foil, plastic wrap, stand, cart, books, book, glasses, handle, backrest, toys, toy, doll, accessories, clothing, swimsuit, dress, skirts, pants, roses, tulips, flowers, plants, leaves, branches, umbrella, umbrellas, hat, changing table, fire place, fireplace, pacifier, refrigerator magnet, urinal

# E ATTRIBUTE CATEGORIES

This section presents the comprehensive attribute categories used in our evaluation framework. The attributes are organized into five main domains: colors, numbers, materials, physical properties, and environmental conditions.

## E.1 COLORS

- **Single Colors:** blue, white, red, black, brown, green, yellow, orange, pink, grey, gray, purple, silver, tan, beige, cream, gold
- **Color Combinations:** black and white, blue and white, black and yellow, green and yellow, brown and white, black and red, black and gray, white and gray
- **Color Descriptors:** light blue, dark red, mint green, colorful, rainbow colored, monochrome, dark, light, color, colored, different colors, colors, browns, whites, rosy, colorfully, red-haired, blonde-haired, ginger, creamy

## E.2 NUMBERS

- **Basic Numbers:** one, two, three, four, five, six, seven
- **Written Numbers:** 2, 3, 25, 50
- **Ordinals:** first, second, third
- **Quantities:** a, another, solo, whole
- **Prices:** 11.98, 10.99

## E.3 MATERIALS

- **Materials:** wooden, wood, metal, plastic, glass, ceramic, concrete, stainless steel, tile, brick, cement, marble, leather, fabric, steel, granite, stone, plywood, paper
- **Surface Qualities and Textures:** striped, polka dot, polka-dotted, polka dotted, tiled, plain, painted, polished, scratched, printed, stripped

### E.4 PHYSICAL PROPERTIES

- **Shapes:** square, round, circular, oval, rectangular, triangular

- **Physical Descriptors:** thick, thin, stuffed, sliced, ripe, unripe, wet, dry, clean, muddy, squares, wedges, opaque, clear, edge, back, duck shaped, fish shaped, horned, antlered

### E.5 ENVIRONMENTAL CONDITIONS

- **Weather:** sunny, snowy, cloudy, overcast, stormy, wet

- **Landscape/Terrain:** grassy, grass covered, snow covered, rocky, sandy, lush, dry, desert, tropical, remote, green, fenced

- **Water Depth:** knee deep, ankle deep

- **Light Conditions:** dim, bright

## F  QUALITATIVE EXAMPLES

In this section, we present qualitative examples from CROSSCHECK, illustrating the variety of object and attribute categories. The dataset covers five object categories – animals, transportation, food, sports, and household items, and five attribute categories – colors, numbers, materials, physical properties, and environmental conditions. Fig. 8 shows qualitative examples from each category in CROSSCHECK.

## G  HUMAN VALIDATION

Annotators assess three components for each sample: conflicting captions (verifying single-element modifications involving plausible, objective properties while avoiding impossible or subjective changes like man-to-woman), questions (confirming clarity, unambiguity, and focus on the changed element), and answers (checking for distinctiveness, objectivity, and visual observability while identifying problematic vague terms like "medium" or "beautiful"). Based on this assessment, annotators provide a single accept/reject decision for each sample. The exact instructions provided to the annotators are shown below, while Fig. 9 depicts a few examples of accepted and rejected samples. Fig. 10 shows examples from the human verification interface. Annotators evaluate each sample using binary accept/reject votes to ensure the benchmark's reliability.

---

**Human validation instructions**

You will evaluate three things for each example:

1. **Conflicting Caption**
   - Did the caption change only one clear attribute or object (the change is marked in bold)?
   - Is the change plausible and objective (e.g., color, number, shape, material, texture)?

2. **Question**
   - Is the question clear and unambiguous?
   - Does it focus on the changed attribute or object?
   - Can it be answered using the given options?

3. **Answers**
   - Are the answers distinct and not synonyms?
   - Are they objective and visual (e.g., colors, numbers, objects)?

**Your task:** For each sample, mark whether it is sensible (Yes/No).

---

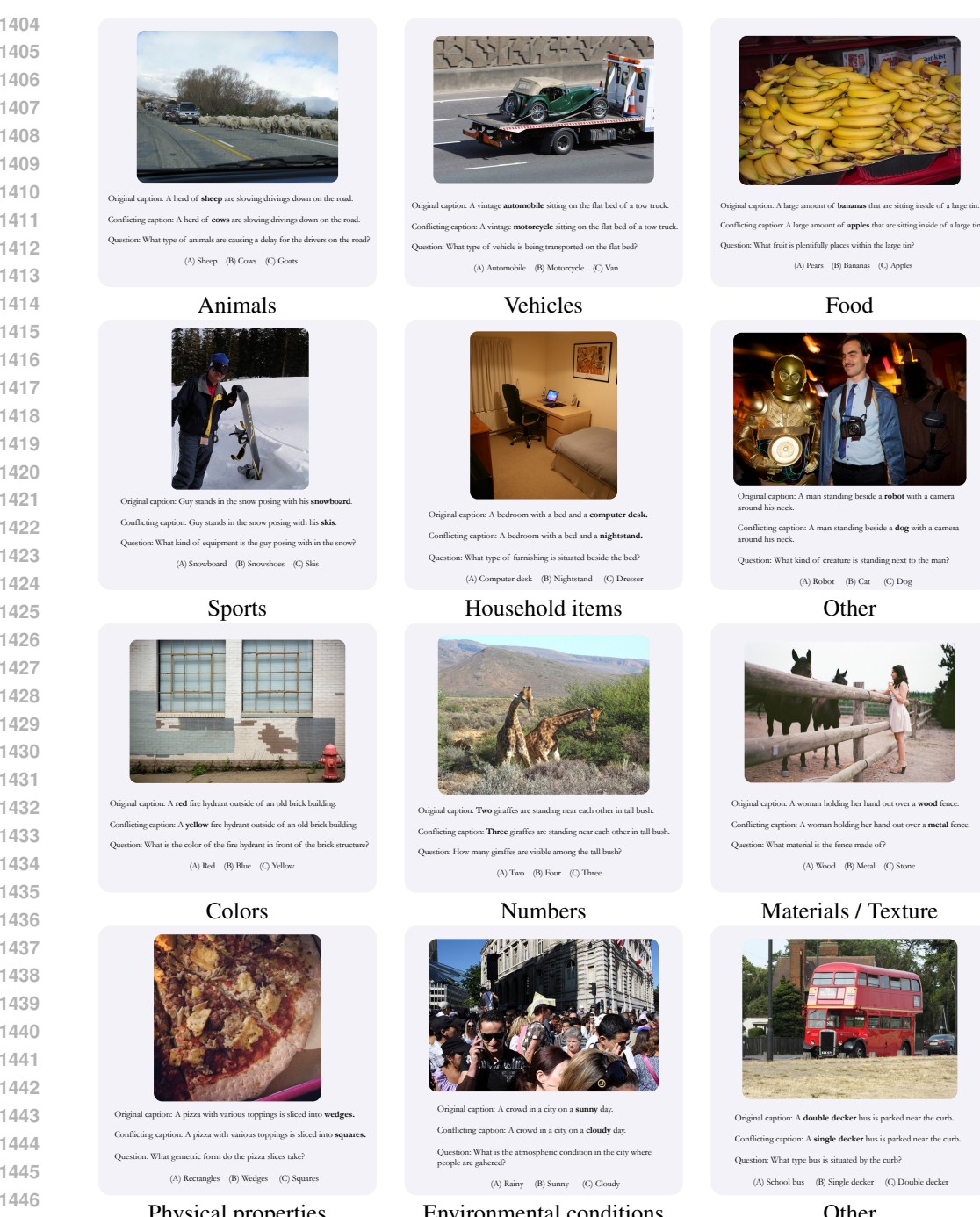

Figure 8: Qualitative examples from CROSSCHECK, illustrating the diversity of object and attribute categories. The dataset spans five object categories (top two rows): animals, transportation, food, sports, and household items, and five attribute categories (bottom two rows): colors, numbers, materials, physical properties, and environmental conditions.

## H   LLM USAGE

Large language models (Claude Sonnet 4 (Anthropic, 2025), GPT-5 (OpenAI, 2025)) were used for manuscript preparation assistance, including text polishing and grammar correction. As detailed in the main paper (§ 3.2 and § 3.3), LLMs (Gemini 2.5 Pro and Gemini 2.5 Flash Lite (Team et al.,

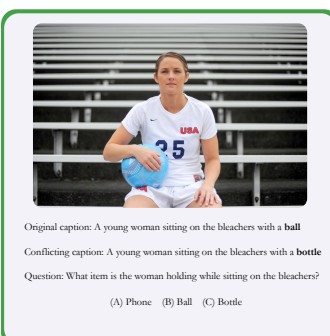 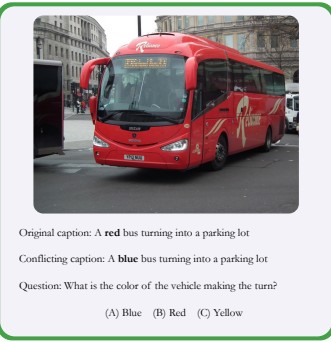 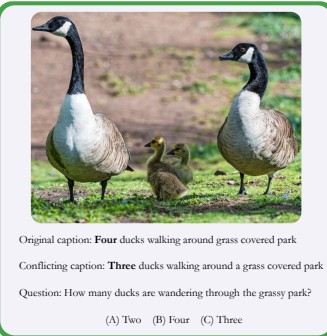

(a) Changed object      (b) Changed attribute (color)      (c) Changed attribute (number)

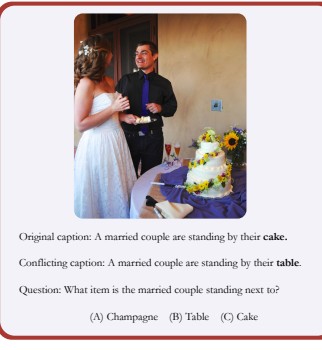 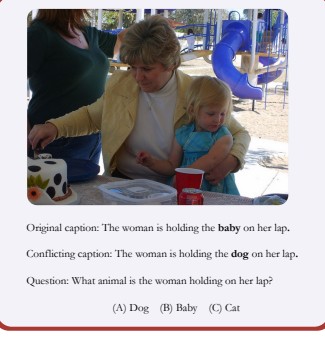 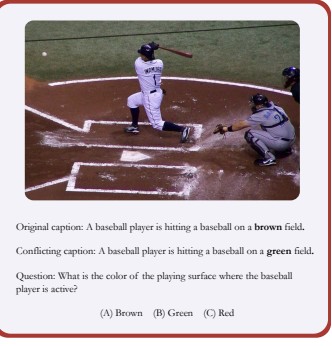

(d) Problematic conflicting word      (e) Problematic question      (f) Problematic answers

Figure 9: Examples of accepted (**top row**) and rejected (**bottom row**) samples during human validation. Positive examples illustrate cases where the conflicting caption, question, and answers are clear and unambiguous. Negative examples highlight typical sources of rejection: (1) conflicting words, e.g., the change is "cake → table" but the image contains both a table and a cake; (2) problematic questions, e.g., the change is "baby → dog" but the question implies an animal; and (3) problematic answers, e.g., the color of the field could be described as being "red" (distractor answer).

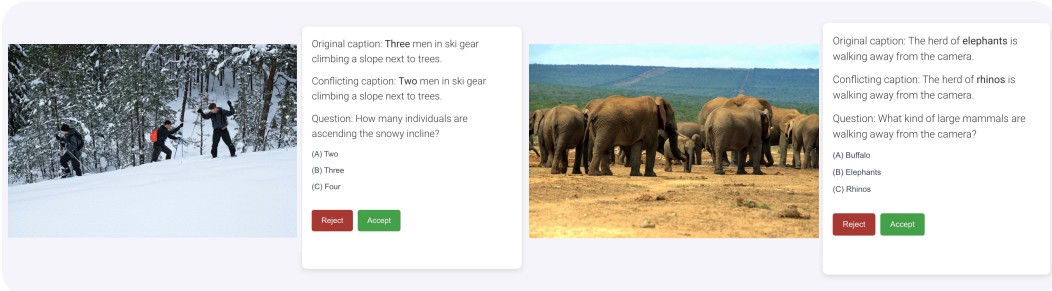

Figure 10: Examples from the human verification interface. Each sample includes an original caption from MS COCO, a conflicting caption that introduces a controlled contradiction, a targeted question designed to test conflict detection, and multiple-choice answers. Annotators use binary accept/reject voting to validate the quality and clarity of each sample, ensuring the reliability of the benchmark's diagnostic test set.

2023)) were used to generate caption modifications, questions targeting the modified content, and answer choices for our dataset, followed by extensive filtering, and human validation of the test set.

## I  BROADER IMPACT

This work contributes to the development of more reliable multimodal AI systems by exposing critical limitations in conflict detection capabilities. Improved conflict detection could enhance AI safety in applications like medical diagnosis, autonomous systems, and content verification. However, the focus on synthetic contradictions may not fully represent the complexity of real-world misinformation or adversarial scenarios. We encourage future work to extend these findings to more diverse contradiction types and real-world deployment contexts.

