# OpenReview forum: "CrossCheck: A Vision-Language Conflict Detection Benchmark"
_ICLR.cc/2026/Conference — ICLR 2026 Conference Withdrawn Submission_

### Official Review · Reviewer_5w3d · 2025-10-17

**Soundness:** 2
**Presentation:** 2
**Contribution:** 2
**Rating:** 2
**Confidence:** 3

**Summary:**

The paper introduces CrossCheck, a benchmark designed to evaluate MM-LLMs on their ability to detect contradictions between images and text. The authors generate controlled conflicts (object-level or attribute-level) using COCO images and LLM-generated captions, then pair them with targeted multiple-choice and open-ended questions. The benchmark contains roughly 15k filtered training samples and a human-verified diagnostic test set. Evaluations show that leading closed-source models (GPT-5, Gemini 2.5 Pro) perform well, while open-source models fail but can be improved substantially through fine-tuning.

**Strengths:**

- This paper tackles a timely topic of cross-modal contradiction detection.

- The paper describes a multi-stage pipeline with automatic and human validation, which is clearly presented and relatively transparent. The benchmark includes both multiple-choice and open-ended tasks, with detailed analyses of modality bias and category-specific weaknesses, showing clear behavioral patterns in models.

- The LoRA experiments are convincing that task-specific data can substantially improve open-source model performance.

**Weaknesses:**

- While positioned as a benchmark for “conflict detection,” the approach essentially reformulates existing COCO data with synthetic contradictions. This resembles prior hallucination or consistency datasets (e.g., POPE, NOPE, HallusionBench) but with minor framing changes. The conceptual contribution feels incremental.

- Most contradictions are generated automatically by an LLM. The resulting text often appears templated and artificial, raising concerns about linguistic realism and whether models are learning to detect genuine conflicts or just distributional artifacts.

- The benchmark only handles object- and attribute-level mismatches. More complex inconsistencies such as spatial, relational, temporal, causal, or narrative conflicts are not covered, which limits generalization and real-world relevance.

- Accuracy is the sole metric. The paper does not analyze confidence, calibration, or detection trade-offs (e.g., false positives on consistent inputs). Thus, the claim of “conflict detection” as a reasoning ability is not strongly supported by the current analysis.

- The LoRA results show large gains, but the paper doesn’t clarify whether these improvements reflect better contradiction reasoning or simply pattern memorization of question templates.

- Duplicated information provided by Figure 1 (right) and Figure 2 (left and middle).

**Questions:**

Please see weaknesses

---

### Official Review · Reviewer_t6ep · 2025-10-27

**Soundness:** 3
**Presentation:** 3
**Contribution:** 2
**Rating:** 4
**Confidence:** 3

**Summary:**

This paper introduces CROSSCHECK, a new benchmark designed to evaluate cross-modal inconsistency detection in multimodal large language models (MLLMs). The benchmark consists of image-caption pairs extracted from MS-COCO, where each sample contains exactly one controlled contradiction between visual and textual content at either the object or attribute level. Both multiple-choice and open-ended question formats are provided. Experiments conducted on 11 closed- and open-source MLLMs (e.g., GPT-5, Gemini 2.5 Pro, LLaVA-1.5, InstructBLIP) show that top closed-source models exceed 85–90% accuracy, while open-source ones lag behind, revealing current performance gaps.

**Strengths:**

**S1.** While some existing benchmarks cover cross-modal inconsistency (see **W1**), the proposed benchmark is among the first to explicitly benchmark cross-modal inconsistency detection in image-caption settings.

**S2.** The proposed benchmark supports both multiple-choice and open-ended question formats, enabling fair evaluation of both discriminative and generative multimodal models. This dual design is particularly suitable for recent MLLMs.

**S3.** All contradictory captions were manually verified for semantic correctness and clarity, ensuring high data quality compared to benchmarks that rely primarily on automatic generation.

**S4.** The paper reports systematic results for 11 MLLMs, including both closed- and open-sources, demonstrating the applicability and reproducibility.

**S5.** The paper is mostly well-written.

**Weaknesses:**

**W1. Novelty**
The proposed task can be regarded as a form of unanswerable question detection, where the image content does not contain sufficient information to answer the given question. From this perspective, the problem formulation overlaps with prior studies on Unanswerable Question Answering and Unsolvable Problem Detection [a-c], which also address the detection of cross-modal inconsistencies between visual and textual inputs. As these works already explore similar settings, the novelty of the proposed benchmark appears somewhat limited.


**W2. Benchmark Scope and Potential Saturation**

**W2-1.** The coverage of the proposed benchmark appears rather limited. Its scope is confined to object- and attribute-level contradictions, which are generated mainly by word substitutions on MS-COCO captions. This design narrows the range of real-world inconsistencies and may fail to test reasoning types that multimodal LLMs usually find difficult, such as geometric, spatial, or contextual reasoning.

The authors argue in the limitation section that spatial relations, temporal sequences, actions/events, and broader contextual contradictions are outside the current scope. However, recent benchmarks explore a much wider range of cross-modal inconsistencies, including relation-level reasoning [a], layout-based multimodal inconsistencies across five categories [c], and false premises, insufficient context, and visual illusions [d,e]. Taken together, these facts suggest that the proposed benchmark is overly narrow compared with the current state of research.

**W2-2.** Consistent with W2-1, several top closed-source models achieve over 85-90% accuracy on the proposed benchmark, while their performance drops significantly on other benchmarks. This gap indicates possible ceiling effects, suggesting that the benchmark may already be easy for modern MLLMs. Overall, while the dataset is useful as a controlled and well-defined testbed, its limited types of contradictions and simplified construction raise questions about its effectiveness as a benchmark for evaluating cross-modal inconsistency in real-world settings.

------
[a] Akter et al., VISREAS: Complex Visual Reasoning with Unanswerable Questions, ACL Findings, 2024.

[b] Miyai et al., Unsolvable Problem Detection: Robust Understanding Evaluation for Large Multimodal Models, ACL, 2025.

[c] Yan et al., Multimodal Inconsistency Reasoning (MMIR): A New Benchmark for Multimodal Reasoning Models, ACL Findings, 2025.

[d] Wang et al., HaloQuest: A Visual Hallucination Dataset for Advancing Multimodal Reasoning, ECCV, 2024.

[e] Guan et al., HALLUSIONBENCH: An Advanced Diagnostic Suite for Entangled Language Hallucination and Visual Illusion in Large Vision-Language Models, CVPR, 2024.

**Questions:**

**For W1:** How do the authors conceptually distinguish the proposed benchmark from prior formulations such as unanswerable question detection or unsolvable problem detection [a–c]?

**For W2-1:** Given the restriction to object/attribute contradictions, please provide evidence that this scope is sufficient for evaluating cross-modal inconsistency; for example, showing that these types constitute a large share of real-world inconsistencies.

**For W2-2:** Several closed-source models score 85–90%. Please supply objective demonstrations that the benchmark is non-trivial; for instance, additional verification showing that other recent stronger models still face significant errors, or cross-benchmark comparisons indicating non-saturation.

---

### Official Review · Reviewer_X6qS · 2025-10-30

**Soundness:** 2
**Presentation:** 2
**Contribution:** 2
**Rating:** 4
**Confidence:** 4

**Summary:**

This paper introduces CROSSCHECK, a new benchmark designed to evaluate multimodal conflict detection, the ability of models to recognize contradictions between images and text. The benchmark pairs COCO images with captions containing controlled object-level and attribute-level conflicts, and includes both multiple-choice and open-ended questions. It consists of an automatically filtered fine-tuning set and a smaller, human-verified diagnostic set. Experiments reveal significant weaknesses in detecting cross-modal contradictions and fine-tuning on CROSSCHECK significantly improves these capabilities.

**Strengths:**

- S1. The problem of image–text conflict is an important topic for evaluating the reliability of LMMs, and the paper has a good focus on this issue.

- S2. The benchmark construction process, including human verification, is carefully and thoughtfully designed.

**Weaknesses:**

- W1. There are already several studies that address this type of conflict problem [1–4]. For example, UPD [1] proposed a task to detect mismatches between images and text, showing the limitations of existing models (especially open-source LMMs) and that fine-tuning can help solve the issue. Therefore, the findings of this paper seems not very novel.


- W2. The open-source LMMs used in the experiments are rather old. Newer versions such as Qwen2.5VL [5], InternVL3.5 [6], and LLaVA-OneVision [7] should be included. Previous work [1] also suggests that larger models perform better at conflict detection, so showing results for models with more than 30B parameters would make the analysis stronger. With the current results, it is difficult to claim that open-source models perform poorly.


- W3. The benchmark is based on COCO, but COCO includes only a limited range of image types. Benchmarks like MMBench use data from various sources and have wider coverage. Although this paper uses many samples, the diversity of image data is still low.


- W4. Fine-tuning might make the model too specialized for this task and reduce its generality. Therefore, it cannot be considered a good solution.




Due to the above reasons, I believe this paper does not reach the acceptance threshold.


[1] Miyai+, Unsolvable Problem Detection: Robust Understanding Evaluation for Large Multimodal Models, ACL 2025


[2] Guo+, UNK-VQA: A Dataset and a Probe Into the Abstention Ability of Multi-Modal Large Models, TPAMI 2024


[3] Akter+, VISREAS: Complex Visual Reasoning with Unanswerable Questions, ACL Findings 2024


[4] Cao+, VisDiaHalBench: A Visual Dialogue Benchmark For Diagnosing Hallucination in Large Vision-Language Models, ACL 2024

[5] Bai+, Qwen2.5-VL Technical Report, arXiv2025

[6] Wang+, InternVL3.5: Advancing Open-Source Multimodal Models in Versatility, Reasoning, and Efficiency, arXiv2025


[7] Li+, LLaVA-OneVision: Easy Visual Task Transfer, TMLR2024

**Questions:**

- I would like to see a clearer distinction from existing similar works.

- It would also be helpful to verify whether the findings remain consistent when using newer or larger LMMs.

- Additionally, I am concerned that fine-tuning may make the model too specialized for this task, reducing its generalization ability.

---

### Official Review · Reviewer_Cuq6 · 2025-11-02

**Soundness:** 3
**Presentation:** 3
**Contribution:** 2
**Rating:** 4
**Confidence:** 4

**Summary:**

This paper introduces CROSSCHECK, a new benchmark for evaluating multimodal large-language models (MM-LLMs) on cross-modal conflict detection. Leveraging MS COCO images, the authors synthetically generate ∼15 K fine-tuned training samples and 1 289 human-verified test cases where a single object or attribute in the caption is altered to contradict the image. Each example is paired with both multiple-choice and open-ended questions designed to pinpoint that controlled inconsistency. The authors evaluate a wide range of closed-source and open-source MM-LLMs, revealing: (1) A stark performance gap (≈ 88% vs. near-0%) between top closed-source models (GPT-5, Gemini 2.5 Pro) and most open-source models on conflict detection. (2) Systematic modality biases: some models over-rely on the image (e.g., GPT-4.1 Mini), others on text (e.g., InternVL). (3) Category-specific weaknesses: environmental attributes are hardest, object categories vary in difficulty (e.g., animals > household items). (4) LoRA fine-tuning on CROSSCHECK dramatically boosts open-source models (e.g., LLaVa-1.5-7b from 0% → 77%).

**Strengths:**

1. This paper is well-written and easy to follow.

2. This paper designs a dataset construction and quality control pipeline, with both multiple-choice and open-ended questions.

3. This paper not only shows the weakness but also provides wa ay to mitigate this weakness.

**Weaknesses:**

1. This paper has a limited scope of contradictions. Only object- and attribute-level mismatches. How well would MM-LLMs handle more complex spatial or action-based conflicts? Besides, only relying on original COCO images and all conflicts are purely textual.

2. The novelty is not strong enough. Prior work (e.g., AutoHallusion, HaloQuest, Koala) already explores synthetic multimodal mismatches via inpainting or targeted perturbations. This paper rehashes the same paradigm using LLMs for caption edits, without offering fundamentally new methodology or insight.

3. No evidence that improvements on CROSSCHECK transfer to external datasets or practical tasks (e.g., medical imaging, document analysis). It will be better to add a new experiment on real-world tasks.

**Questions:**

1. In Table 3, why does the performance of mPLUG-Owl-1 on no conflict also improve (from 31.62 to 63.13)?

2. No evidence that improvements on CROSSCHECK transfer to external datasets or practical tasks (e.g., medical imaging, document analysis). It will be better to add a new experiment on real-world tasks.

3. Broader conflict types, editing methods, and domain diversity are helpful to improve the quality of this paper.

---

### Note · Authors · 2025-11-14

I have read and agree with the venue's withdrawal policy on behalf of myself and my co-authors.